

# Measuring and Characterizing Community Recovery to Earthquake: the Case of 2008 Wenchuan Earthquake, China

**Jie Liu[1], Zhenwu Shi[1], Di Lu[2] and Yongliang Wang[3]**

[1]School of Civil Engineering, Northeast Forestry University, Hexing Road 26, Xiang Fang District, Harbin, China.

[2]Department of Economics and Management, Harbin Institute of Technology, No. 92 Xidazhi Street, Harbin, China.

[3]Harbin Power System Engineering & Research Institute CO., LTD, No.1 Xusheng Street, Harbin, China.

*Correspondence to:* J. Liu (liujie198643@163.com)

**Abstract.** Our world is prone to more frequent, deadly and costly earthquake disasters, which are
increasingly uncertain and complex due to the rapid environmental and socio-economic changes
occurring at multiple scales. There is an urgent need to recover rapidly and effectively for community
after earthquake disasters. To enhance community recovery, it is necessary to have a good initial
understanding of what it is, its determinants and how it can be measured, maintained and improved. So
this article proposes the concept of community recovery as the capacity to recover and rebuild after the
earthquake disasters by considering the original perspective of recovery. And we develop a new
quantitative approach to measure community recovery to earthquake from four dimensions (population,
economic, building, and infrastructure) by extending the concepts of recovery triangle. Taking the
community of Wenchuan as the example to test our mathematical model and compare different recovery
levels of four dimensions under the situation of Wenchuan Earthquake, the results can help the policy
makers identify the low-recovery dimensions of Wenchuan to enhance post-disaster recovery and
reconstruction efforts, and address the vital importance of local government in improving the
post-disaster recovery.
**1 Introduction**
The damaging earthquake risk of cities as the biggest risk of all natural disasters has specifically
increased over the years due to the increasing complexities in urban environments and a high
concentrated urbanization in seismic risk-prone areas. The growing large-scale devastating effects
caused by recent catastrophic earthquakes (e.g. 15 August 2007, Peru; 12 May 2008, Wenchuan, China;
12 January, 2010, Haiti; 11 March 2011, Honshu Island, Japan) have attracted the attention of the policy
makers to formulate effective risk prevention policies. The earthquake risk depends on the seismic
hazard, but it is more dependent on the inherent properties of the communities which is compounded by
the vulnerability, adaptation and resilience. Above all of these inherent properties, resilience is



interpreted to be the central component of disaster risk reduction, which is used to bridge the two other properties together. Some researchers asserted that a disaster-resilient community has the ability to cope with the disaster strikes, and improve its inherent genetic or behavioral characteristics to better adapt to disasters rather than regain pre-disaster levels of vulnerability (Mooney 2009). So policymakers have called for concerted efforts to build "earthquake-resilience community" for the purpose of finding the new stable states and rebuilding a safer community in the historically experienced deleterious earthquake disasters (Alesch 2009). The definition of resilience is the ability that is exposed to seismic hazards to resist, absorb, accommodate and recover from seismic hazards quickly and efficiently, which is divided by some scholars into during-disaster resistance, short-term post-disaster recovery, and long-term post-disaster trans-formative (UN/ISDR 2010). Recovery represents a fundamental dimension of disaster resilience, includes both the possibilities o return to normal, that is, pre-disaster condition, and alternatively, to be rebuilt or transformed to a completely different status. So reconstruction, restoration, rehabilitation and post-disaster redevelopment are all considered to be the parts of the recovery process, yet it is widely acknowledged to be the final phase of the disaster life cycle (Tierney et al. 2001; NRC 2006; Peacock et al. 2008; Olshansky and Chang 2009).

In academia, recovery has traditionally taken on a more outcome-oriented conceptualization, with emphasis on the physical aspect as seen in early studies (Haas et al. 1977). Researchers like Nigg then began to point out that recovery should be conceptualized as a social process that "begins before a disaster occurs and encompasses decision-making concerning emergency response, restoration, and reconstruction activities following the disaster" (Nigg 1995). Some other scholars have suggested that recovery can be defined as the "process by which a community has experienced a structural failure of this sort to reestablish a routine, organized, institutionalized mode of adaptation to its post-impact environment" since the disaster was often seen as a failure of social structure (Bates and Gillis Peacock 1989). These changes in the definition to reflect the shifts in conceptualizing disaster recovery in the last few decades from a linear, static issue focused on the physical aspects referred to a specific set of stages, to a dynamic, interactive, multi-dimensional decision-making process, including the 'reconstructing, and remodeling of the natural and social-economic environment by pre-disaster planning and post-disaster actions' (Smith and Wenger 2007). And the researches surrounding "disaster recovery" have attracted more and more attention in recent years. Definitions of this term vary in the literature, which are commonly used in the sense of 'returning to pre-disaster conditions', or 'reaching a new stable state that may be different from either of these' (Quarantelli 1999). The new National Disaster Recovery Framework developed by FEMA in 2011(FEMA 2011) define recovery as "those capabilities necessary to assist communities affected by an incident to recover effectively, including, but not limited to, rebuilding infrastructure systems, providing adequate interim and long-term housing for survivors; restoring health, social, and community services; promoting economic development; and restoring natural and cultural resources". And community recovery emerges "as the outcome of several sets of activities: restoring basic services to acceptable levels, replacing infrastructure capacity that is





damaged or destroyed, rebuilding or replacing critical social or economic elements of the community
that are damaged or lost, and establishing or reestablishing relationships and linkages among critical
elements of the community" (Alesch et al. 2009).
In recent years, much of the current disaster literature provides two major perspectives and
interpretations to measure recovery: (i) returning to pre-disaster situations; and (ii) obtaining a new
normal conditions (Chang et al. 2011). The first perspective and interpretation is conceptually based on
the comparison of the community conditions before the disaster and after the recovery process, and it
emphasizing on the rebounding as quickly as possible (Wildavsky 1991; Sherrieb et al. 2010). In this
regard, the pre-disaster situations are considered to be the normal state. The rapid recovery process is
designed to minimize losses caused by disasters (Alesch et al. 2001). The second perspective and
interpretation highlights how there is a new normal state after a disaster (Alesch et al. 2009; Chang et al.
2010). However, the 'new normal state' is more applicable to post-disaster attitudes and behavior of
human, showing the evolution of the collective psychology, than it is to physical recovery. Beside that,
some recovery indexes have been designed to track the recovery progress, such as the Social
Vulnerability Index proposed by Cutter and Finch (2008), Spatial Recovery Index (SRI) proposed by
Ward et al. (2010) and so on. These recovery indexes resonate with the fine view of the bouncing back
method in as much as these dimensions are critical to understand the post-disaster improved situations.
Nowadays, the research of disaster recovery is in the initial stage, more key research questions need to
be resolved: Why do some communities recover more quickly and successfully than others? Is there a
timetable for recovery? How does the recovery trajectory of communities differ by type and magnitude
of the hazard event, conditions of initial damage, characteristics of the community, and decisions made
over the course of reconstruction and recovery? How do different types of assistance and recovery
resources affect recovery? What types of decisions and strategies are most critical to recovery? How do
disasters affect communities over the long term? In the past studies, the idea of post-disaster
improvement is preferred by many scholars to the idea of bringing back to or regaining the pre-disaster
normality, especially when the disasters are occurring in developing countries, while the concepts and
practices of sustainable development and risk reduction are being integrated into disaster recovery
processes. The concept of disaster recovery is recognized as ordered, knowable, and predicable, for the
emphasis is mainly focus on the building environment. However, later studies have shown that the
recovery process does not follow a predictable timeline, and that the recovery process is increasingly to
multi-dimensional, including both physical (economic) and social-psychological aspects. The
determinants of disaster recovery are many, include socioeconomic status and development trends,
structural change and adaptation, disaster impacts and disruptions, post-disaster response efforts,
informal and formal external assistance (governmental and institutional capacity), and
macro-socioeconomic or program/policy changes. So the measurement of disaster recovery is a
complex construct, a recurrent problem is the lack of a simple, feasible and effective measurement of
disaster recovery. So in this paper, we proposed a new, practical method for measuring and




characterizing community recovery to earthquake in four dimensions, and applied it to Wenchuan
Community. The final products of our research provide insights for decision-makers to acknowledge
and understand the differential levels of community recovery in these four dimensions, in order to
maximize the overall post-disaster community recovery by prioritizing efforts, and formulating effective,
operational and valuable reconstruction strategies and policies.
**2 Study Area**
The Wenchuan Community (31°East, 103.4°North) in Sichuan Province of China was hit by a
magnitude 8.0 Ms (the surface-wave magnitude) and 7.9 Mw earthquake (Wenchuan Earthquake)
(Figure 1) at 14:28:04 CST (China Standard Time) on May 12, 2008. The Epicentral intensity of this
earthquake was up to 11 degrees, and the areas directly devastated by this earthquake were as large as
100,000 square kilometers. Wenchuan Earthquake is the most destructive and widespread earthquake
since the founding of the People's Republic of China, which affected more than half of China and other
Asian countries and regions. Up to September 18, 2008, the Wenchuan Earthquake caused 69,227
people dead, 374,643 injured, and 17,923 missing. Direct economic losses reached 845.2 billion yuan
($ 133.2 billion). The Wenchuan Community as the epicenter of Wenchuan earthquake was the hardest
hit (Figure 2b). In Wenchuan Community, this earthquake left 15,941 people dead, 34,583 injured, and
7,930 people have been listed as missing. The Wenchuan Community was razed by this earthquake: all
infrastructures were completely destroyed, most buildings were severely damaged, many economic
sectors such as industry, commerce and tourism were suffered heavy losses (64.3 billion yuan ($ 10.1
billion) in direct economic losses).

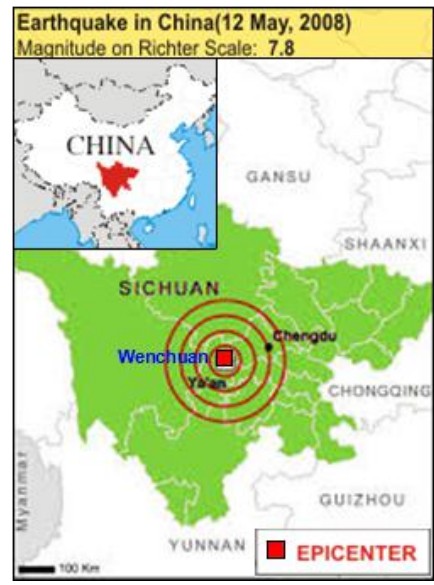

**Figure 1.** Location of Wenchuan Earthquake






After Wenchuan Earthquake, Chinese Central Government commanded a large number of rescuers
(including firefighters, special police, volunteers and humanitarian relief experts) from all over China
and around the world to take emergency response measures. On June 8, 2008, "Regulations on
Post-Wenchuan Earthquake Rehabilitation and Reconstruction" was promulgated, and the Chinese
government announced to invest 1 trillion yuan ($157.7 billion) to rebuild the affected areas over the
next 3 years. In the rebuilding and recovery processes, with the principle of "one province helps one
severely affected communities", 19 provinces and cities (e.g. Guangdong, Jiangsu, Shanghai, Shandong,
Zhejiang, Beijing, Liaoning, Henan, Hebei, Shanxi, Fujian, Huan, Hubei, Anhui, Tianjin, Heilongjiang,
Chonging, Jiangxi, Jilin) supported the reconstruction of 18 worst-hit communities (e.g. Wenchuan,
Qingchuan, Beichuan, Mianzhu, and so on) for three years. Just forced on the Wenchuan Community,
the reconstruction projects of the national plan are more than 4,000, with the total investment of 40
billion yuan ($ 6.3 billion) from 2008 to 2011. On the third anniversary of Wenchuan Earthquake (May
12, 2011), the reconstruction of the Wenchuan Community is completed, and the Wenchuan Community
is from ruins to prosperity (Figure 2c).

140

| The aerial image of the Wenchuan Community before Wenchuan Earthquake | The aerial image of the Wenchuan Community after Wenchuan Earthquake | The aerial image of the reconstructed Wenchuan Community |
|---|---|---|

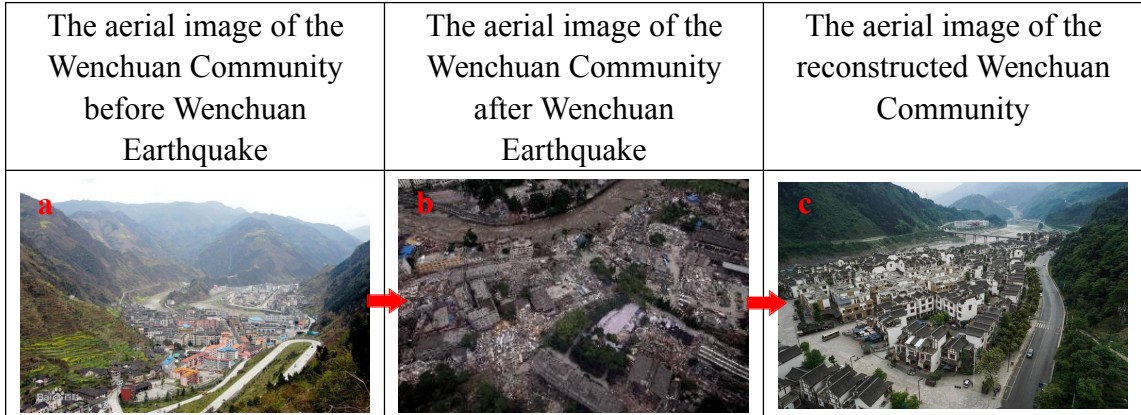

**Figure 2.** The development process of the Wenchuan Community in, during, and after Wenchuan
Earthquake (from May 12, 2008 to May 12, 2011)

## 3 Data and Methods

### 3.1 Data Sources

Data of the detail reconstruction or recovery processes of the Wenchuan Community after the
earthquake including population, economy, building and infrastructure are mainly obtained from the
reports on the work of the Wenchuan government from 2008 to 2014. Data of the recovery process and
status of the affected people are gotten by the random interview of 1000 affected families from all



resettlement sites. Other statistics and description data are gathered by combining different sources (e.g.,
official statistical yearbooks, newspapers and media reports) following the Wenchuan Earthquake. And
the local information of the reconstruction processes of buildings and infrastructure of Wenchuan
Community, which are obtained by field surveys and interviews. After the earthquake, the government
made every effort to restore infrastructure services of the affected areas, and the emergency water
supply, telecommunications, electricity, and roads were recovered respectively on May 13, May 15,
May 17, and August 12, 2008. With regarding to repair and rebuild the earthquake-affected buildings,
501 reconstruction projects with the total investment of 22.177 billion yuan ($ 3.5 billion)are completed
in Wenchuan Community. From 2008 to 2011, reconstruction projects had been completed by 19%,
53%, and 94.7% in each year. In 2012, all of these 501 reconstruction projects were completed. These
all data were entered into a computerized database. This database was an important source of
information for measuring the recovery of the Wenchuan Community to the earthquake.

**3.2 Defining the concept of community recovery to earthquake**

The researches contain many major conceptual and measurement approaches to define and measure
community recovery. Community recovery, as the final phase of the disaster life cycle, continues
beyond emergency response, that might be taken in the immediate aftermath of a disruption until
returning to pre-disaster normality or transforming to a new stable state. This phase may take days,
months, even years, to accomplish; thus, requiring long-term planning. The recovery is a dynamic,
complex and challenging process that involves all sectors of a community, comprised of the impact of
disasters, households, business, buildings, as well the lifeline system (Miles and Chang, 2007). In many
cases, it is not even clear if and when recovery has been achieved because of varying stakeholder goals
for the community, for example with some wanting it returned to its pre-disaster status and others
wanting it to undergo change to realize a vision in which advances are made in risk reduction and other
areas. But most of all, the decision-makers of local governments mainly through improving the speed of
the recovery process to restore the operation of the interrupted business, and to rebuild damaged
infrastructure to allow the restarting of normal activities (Alesch et al. 2001). So the speed of the
recovery process can be defined as the key indicator of measuring the community recovery in much
disaster literature. In this paper, we define the concept of community recovery as the capacity of a
community to recover and rebuild itself rapidly to an acceptable level of functioning and structure
following the earthquake disaster occurs (Figure 3).





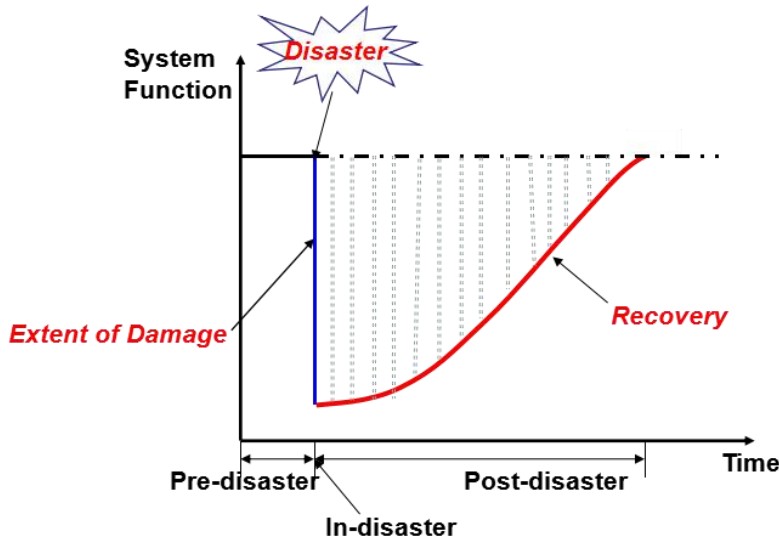


**Figure 3.** The the concept of community recovery


Since recovery begins when a community "repairs or develops social, political, and economic processes
that enable it to function in the new context within which it finds itself" (Alesch et al. 2009). When a
devastating earthquake hits a community, people are injured or killed, economy interruption begins,
buildings are collapsed, and infrastructures are disrupted. The ability of the community to carry out
recovery activities to minimize the immediate impacts caused by an earthquake. According to the
characteristics of earthquake disaster, and in order to better interpret all aspects of recovery of the
community, the community recovery in this paper is divided into four dimensions (such as population,
economy, buildings, and infrastructure):
(1) Population recovery
Earthquake disasters are becoming more complex and uncertain in recent years as a result of the
increasing populations living in seismic areas, which is considered to be exposed to a relatively high
degree of earthquake risk. So this would increase the population affected by earthquake disasters, which
in further can increase the pre-disaster extent of casualties. On the contrary, the trend of rapid
urbanization could induce a future of increased post-disaster population recovery (e.g. the growth rate
can be described as the population recovery in Figure 3). And benefits and restoration efforts are
distributed unequally in the recovery process amongst different sub-populations according to their
geographic locations, socioeconomic status, and different reconstruction programs. So in this paper, the
population recovery is measured based on the index of the average growth rate of the proportion of the
recovered population (e.g. the injured people were treated, the homeless people were placed) in the total
affected population after an earthquake disaster.
(2) Economic recovery

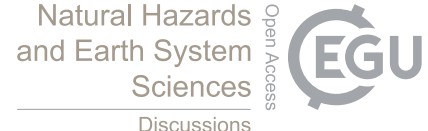

Economic recovery as a promoter of recovery, refers to making the best of the internal and external resources that are available to speed recovery to return to a previous level of economic function at a given point in post-disaster time. The local economic status determines how rapidly a community can recover from such earthquake disasters (Lee 2014; Anne and Adam 2011). Continuation of trends that have been concentrating on the increased significantly economic damage (EM-DAT 2012), while increasing economic development has increased economic vulnerability to earthquake disasters, and in turn a strong and diverse regional economy have direct influence on the recovery capacity to earthquake disasters (e.g. the growth rate can be described as the economic recovery in Figure 4). So in this paper, the economic recovery is measured based on the index of the average growth rate of gross domestic product (GDP) of the affected area after an earthquake disaster.

(3) Building recovery

Building recovery refers to the capacity of a community for post-disaster building reconstruction and retrofitting, which are often amenable to taking on board resilient technologies, given that they have witnessed the effects of the initial threat. The resilient buildings can adjust to certain changes in conditions to counteract damaging structural reactions in response to an seismic hazard. Buildings built with adequate consideration of the earthquake effects that are appropriate for their location dominate the exposure to earthquakes. And the application of earthquake-resistant building codes can make buildings not be seismically vulnerable by helping to prevent or minimize damage to the built environment during earthquake disasters. High-level building recovery is addressed in rebuilding and retrofitting these earthquake resistant buildings (e.g. the rebuild rate can be described as the building recovery in Figure 4), which helps to build-in recovery and provide enhanced safety built environment for community. So in this paper, the building recovery is measured based on the index of average rebuilding rate of the collapsed buildings of the affected area after an earthquake disaster.

(4) Infrastructure recovery

Infrastructure recovery is the judgment to characterize the ability of the key infrastructure which is threatened and disrupted by the earthquake disasters to recover function to the extent possible in post-disaster time. The disruption of the infrastructure system in a major earthquake disaster as the indirect economic damage of a community, suggests whether such community to be resilient, to what extent. A resilient infrastructure system must be designed to continue functioning under extreme seismic hazard conditions, which is a priority goal for earthquake-resilient communities. The capacity for critical infrastructure to quickly restore services following an earthquake determines how rapidly communities can recover from such disasters. Many researches rank the availability of electricity, roads, telecommunications, and water supply as the top four critical infrastructure or lifeline systems that need to function following an earthquake (O'Rourke 2009). A high rate of infrastructure deterioration may be due to the poor quality, the aged equipment, and the highly exposed locations, while the development of the infrastructure system is identified as a strategic priority to be essential to increase the recovery of infrastructure (e.g. the recovered rate can be described as the infrastructure recovery in Figure 4). So in




this paper, the infrastructure recovery is measured based on the index of the average recovered rate of
the disrupted infrastructures of the affected area after an earthquake disaster.

**3.3 Measuring the community recovery to earthquake**

The approach taken in this paper for measuring community recovery is based upon the concept of the
disaster recovery triangle. Originally introduced by Bruneauetal, and extended by Zobel, the disaster
recovery is calculated by two factors: robustness (the strength of the system, measured by its ability to
resist the impact of a disaster event, in terms of the extent of damage suffered be cause of the event),
and rapidity (the rate at which a system is able to recover to an acceptable level of functionality). And
the disaster recovery triangle (in the form of the area above the quality curve) represented the
relationship between these two factors. So for example, the area 1 of the triangle (calculated by the
product of the extent of damage and the time needed to recover normal operations) can be interpreted to
assess the recovery of community 1 in Figure 5. However, in our opinion, using the disaster recovery
triangle to measure the recovery is not so accurate. Firstly, robustness as one factor of this triangle,
which addressed the ability to resist the disaster, is generally considered to be the extent of damage of
the community. Secondly, the disaster recovery triangle can not be accurately used by decision makers
to compare the recovery of different communities. For example, in Figure 4, if the initial extent of
damage ($X_2$) is the same, the size of the area (Area $2_{(a)}$<Area $2_{(b)}$) can represent the degree of the
recovery (Recovery $2_{(a)}$>Recovery $2_{(b)}$) of the communities (Community $2_{(a)}$, Community $2_{(b)}$). But if the
initial extent of damage ($X_1$<$X_2$) is different, the size of the area (Area 1<Area $2_{(a)}$) can't represent the
degree of the recovery (Recovery 1<Recovery $2_{(a)}$) of the communities (Community 1, Community $2_{(a)}$).
The smaller size of the area 1 is due to the less extent of damage, but the recovery curve is not very high.

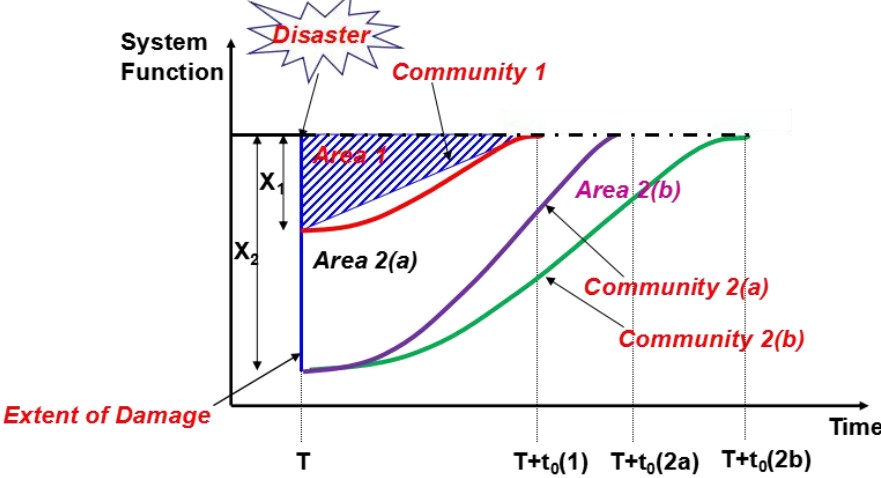

**Figure 4.** The concept of the recovery triangle





Therefore, to compare the recovery of different community, this paper extends the original concept of recovery triangle and proposes a new recovery measurement to fit this paradigm. We use the recover rate to measure community recovery (see in Figure 5). However, the slope of the curve is different at each time point, and not a constant. For the purpose of facilitating the calculation, we use the average linear rate to substitute the curve rate. We let X as the extent of damage to represent the percentage of functionality lost, and we let $t_0(1)$ and $t_0(2)$ represent the time needed to recover normal operations. Based on the principle of the equal area, the community recovery (R) can be measured as the slope of the average linear rate ($\alpha$ is the angle of this line). The entire processes of calculating are as follows:

$$\text{Area 1=Area 2=}\frac{X \times t_0(2)}{2} \rightarrow t_0(2)=\frac{2 \times Area1}{X} \rightarrow R=tan\alpha=\frac{X}{t_0(2)} \qquad (1)$$

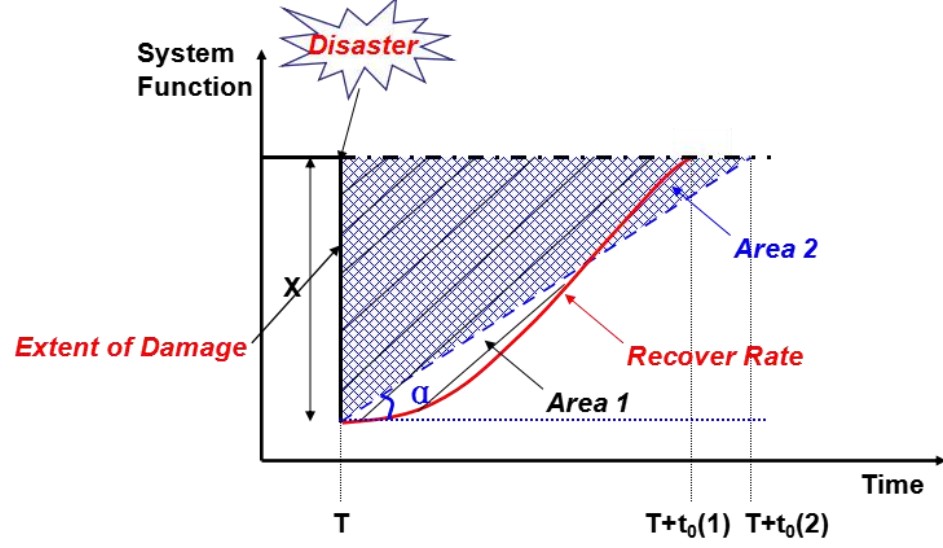

**Figure 5.** The measurement extended from the concept of recovery triangle

## 4 Results

In the result of our study, with the community recovery measuring approach proposed in 3.3 (formula 1), we calculate the recovery scores of Wenchuan Community in four dimensions (population recovery, economic recovery, building recovery and infrastructure recovery), respectively (seen in Figure 6). And three levels (low-recovery, medium-recovery, high-recovery) with the recovery scores of [0-0.577]($\alpha$=0º-30º), [0.577-1.732]($\alpha$=30º-60º), [1.732-+∞] ($\alpha$=60º-90º) are adopted in this study to assess the degree of recovery. The results suggest that the recovery score of economy ($R_{economy}$=1.15) is minimum, and the recovery score of infrastructure ($R_{infrastructure}$=135.19) is maximum. And the economic recovery of Wenhuan which belongs to the medium-recovery level, the population, buildings and





infrastructure recovery belongs to the high-recovery. Based on the definition of community recovery
proposed in this paper, as the capacity of a community to recover and rebuild itself rapidly to an
acceptable level of functioning and structure following the earthquake disasters occur, four key
parameters need to be set: the percentage of functionality lost (X), the initial pre-disaster status, the
acceptable post-disaster level and the recovery time period. The percentage of functionality lost (X) is
classified into four levels, corresponded to low [0%-25%], medium [25%-50%], high [50%-75%] and
extremely-high [75%-100%] level according to the extent of damage. Due to the time of the Wenchuan
Earthquake occurred (May 12, 2008) and the availability of data, we set the status of these four
dimensions at the beginning of 2008 as the initial pre-disaster status. And with reference to the
characteristics of these four dimensions, we use the average growth rate to determine the acceptable
post-disaster level in measuring population recovery, economic recovery, and use the initial pre-disaster
status as the acceptable post-disaster level in measuring the building recovery and infrastructure
recovery. According to National Research Council (2011), the recovery and reconstruction can be
divided into 6 time periods: immediate (< 72 hours), emergency (3-7 days), the recovery focus on very
Short-run (7-30 days), short-run (1-6 months), medium-run (6 months-1 year) and long-run (> 1 year).
The data used to measure the four dimensions of the community recovery are all standardized (by
dimensional analysis, a dimensionless quantity is a quantity without an associated physical dimension)
to eliminate the impact of the different unit of each parameter.

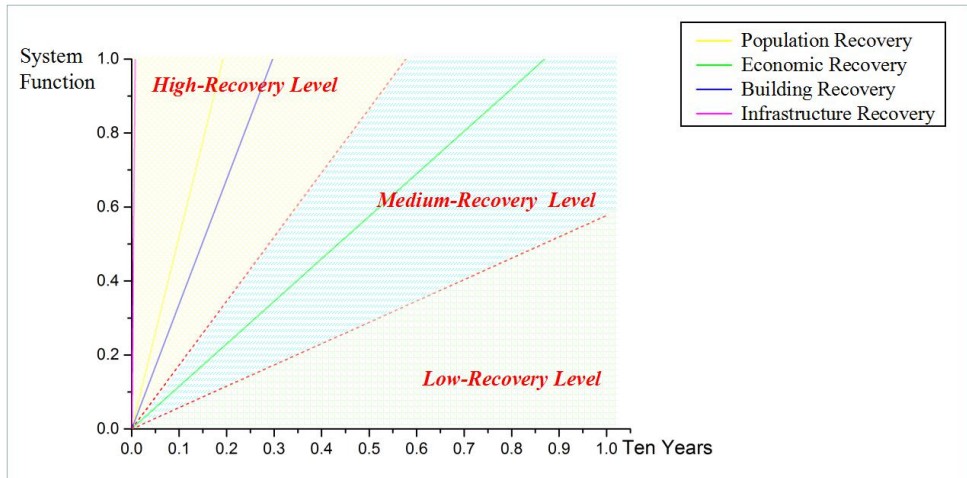

**Figure 6.** The recovery scores of Wenchuan to earthquake in four dimensions

**4.1 Analysis of the population recovery of Wenchuan**

In the result of our study, the recovery process and score of population of Wenchuan is showed in Figure
7. By setting the total affected population of Wenchuan as the initial pre-disaster status, and all of these



affected population return to normal life (e.g. the injured people were treated, the homeless people were
placed) as the acceptable post-disaster level (black dotted line in Figure 7), the population of Wenchuan
recovered in less than three months (blue line in Figure 7), and the population recovery score of
Wehchuan $R_{population}$ is 98.46, which belongs to the high-recovery level, suggesting that the population of
Wenchuan completely recovered from negative effects of earthquake disaster in the short-run time
period. The high-recovery level of population in the process of the post-disaster reconstruction is mainly
due to the rescue principle of the Chinese Central Government that life is of top priority to make the
effective emergency rescue measures. Within 24 hours after the Wenchuan Earthquake occurred, more
than 20,000 soldiers of People's Liberation Army, and 70 medical teams were sent to search and rescue
4,130 wounded, and evacuate more than 3 million affected people. About 1.2 million relief tents,
stretchers and other equipment, more than 800 tons of military food and supplies, 6380 tons of fuel were
transported to the affected area. And 10 settlement sites along the Minjiang River were built around
Wenchuan Community, the remote sensing image of these settlements are showed in Figure 8. The
largest resettlement site is located in Yanmen Township of Wenchuan Community, which covers an area
of about 240 mu. There are more than 2,800 active board houses, which can resettle more than 10,000
affected people.

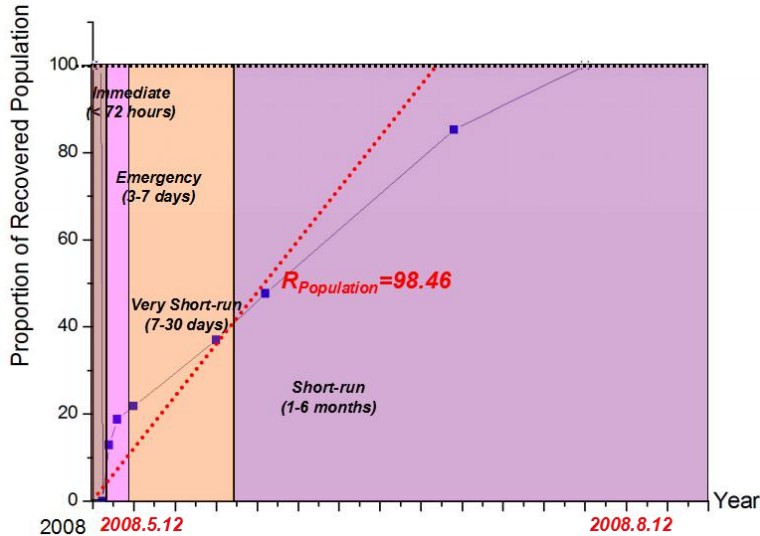

**Figure 7.** The recovery process and score of population of Wenchuan





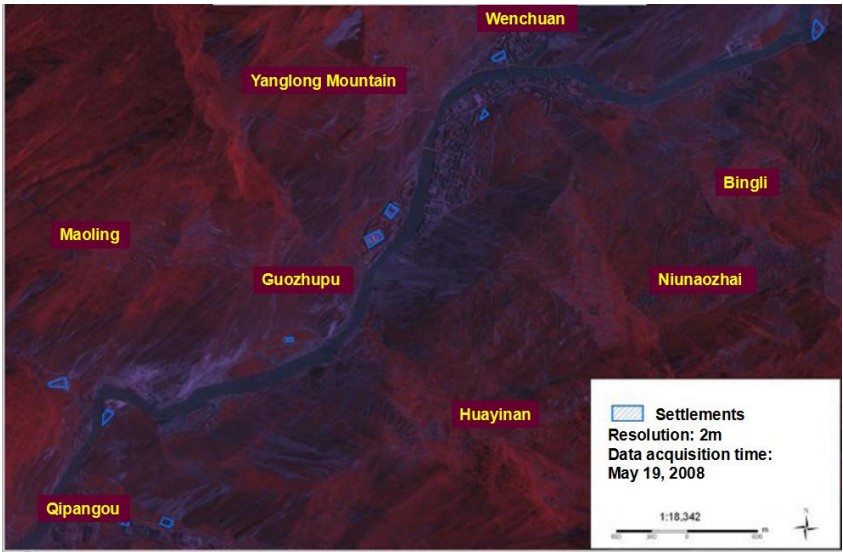


**Figure 8.** The remote sensing image of the settlements of Wenchuan

**4.2 Analysis of the economic recovery of Wenchuan**

The economic recovery pertains to ways for post-disaster economic activities to repair and recover
rapidly (Tierney and Bruneau 2007; Rose 2007). Figure 9 can be interpreted as the economic recovery
process and score of Wenchuan. As set the GDP of Wenchuan at the beginning of 2008 to be the initial
pre-disaster status, the GDP of Wenchuan is only 47.53% of the initial pre-disaster status after Wechuan
Earthquake. During the ground shaking, nearly all property damage occurred immediately. The result
can pinpointed that the economy of Wenchuan is medium extent damage after Wenchuan Earthquake.
The main reasons are the rapid urbanization and the increasing economic development, which
emphasized the significantly increased economic exposure and the economic effects (EMDAT 2012;
World Bank and United Nations 2010). Due to the dynamic characteristics of the economic recovery, we
set the average GDP growth rate of Wenchuan (14.4%) before the earthquake as the acceptable
post-disaster level (black dotted line in Figure 9), and the GDP of Wenchuan have not been recovered
before 2015 (blue line in Figure 9), so we use the average GDP growth rate of Wenchuan (25.2%) after
the earthquake (2008-2015) to forecast the GDP of Wechuan in the future, and the economy of
Wenchuan will recover in 2018 as the long-run time period. The economic recovery score of Wehchuan
$R_{economy}$ is 1.15, which belongs to the medium-recovery level, and is least recovery of these all four
dimensions. Some economic characteristics (a lack of diversified manufacturing and services, a
dependence on specialized entitlements, fragile industrial production chains, low-income settlements,
limited access to economic resources) of Wenchuan contribute to such a long recovery process of the
economy. Aiming to improve the economic recovery to earthquake, built-in a strong and diverse





regional economy will be the most effective scenario. The resilient-economy is not merely make the best of the resources available to return to a previous level of economic function rapidly after the earthquake disasters, but also to increase the capacity of the economic support mechanisms in order to keep the built environment operational and adaptable with the support of post-disaster recovery activities (including contextualizing local economic conditions and prioritizing development projects).

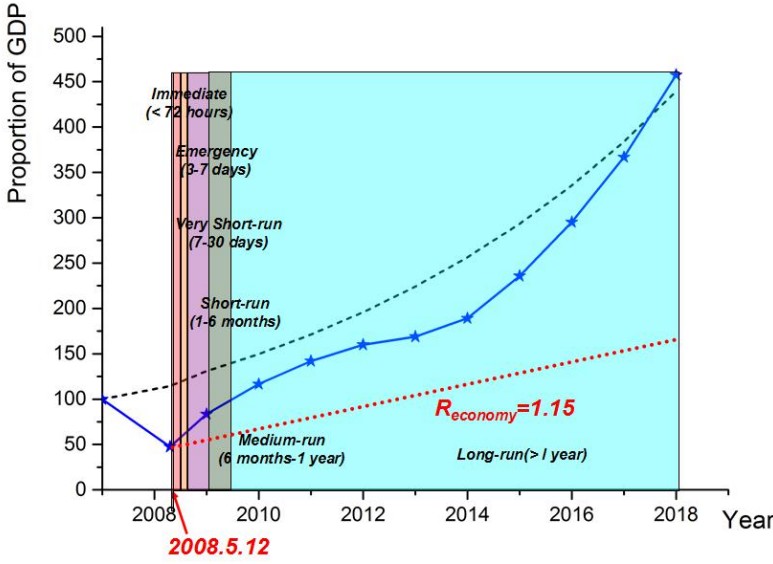

**Figure 9.** The recovery process and score of economy of Wenchuan

## 4.3 Analysis of the building recovery of Wenchuan

Buildings built without adequate consideration of the earthquake effects weaken the community recovery to earthquake. The resulting illustrates that the building recovery process and score of Wenchuan in Figure 10. The total amount of buildings of Wenchuan at the beginning of 2008 can be set as the initial pre-disaster status, and most of these buildings are collapsed in Wenchuan Earthquake, which can be interpreted that the extremely-high extent of damage of buildings with the weakest capacity to resist Wenchuan Earthquake. The low-quality building stock and lack of the earthquake-resistant building codes are the directly and important influencing factor of the extremely-high extent of damage (Jie and Shaoyu 2015). By setting the initial pre-disaster status of buildings as the acceptable post-disaster level (black dotted line in Figure 10), the reconstruction process of buildings of Wenchuan is completed in 2012 as the long-run time period (blue line in Figure 10), and the recovery score of buildings $R_{buildings}$ is 3.37, which belongs to the high-recovery level. According to the guidelines of the central government and heavy financial support ($ 3.5 billion), the local government is almost equivalent to build a "new" Wenchuan Community just over three years,



which highlights the extremely high building recovery of Wenchuan. In Wenchuan Earthquake, the poor
quality of building stock is the key factor responsible for the buildings to be seismically vulnerable. The
new buildings are designed and built with the application of current high seismic design standards,
which can support recovery by helping the built environment prevent or minimize damage during
earthquake disasters.

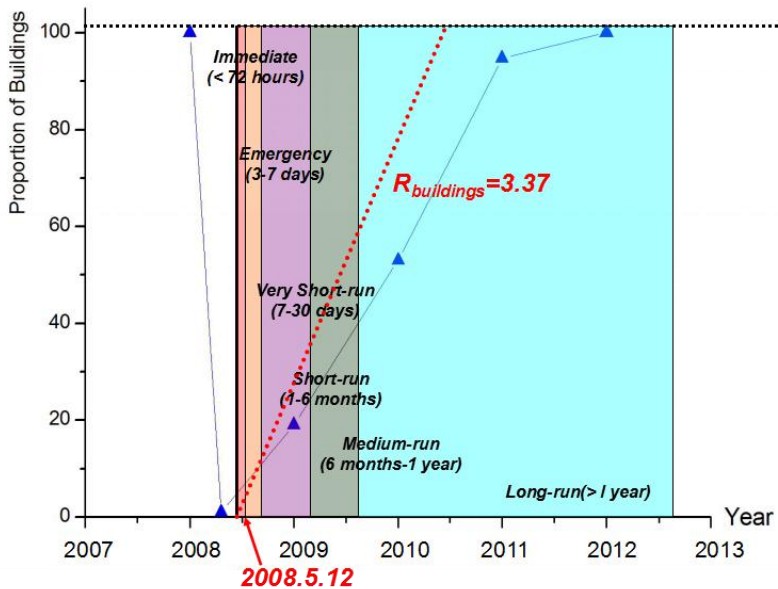

**Figure 10.** The recovery process and score of buildings of Wenchuan

**4.4 Analysis of the infrastructure recovery of Wenchuan**

From Figure 11, we can conclude that infrastructure recovery process and score of Wenchuan. We set
the total amount of infrastructure of Wenchuan at the beginning of 2008 to be the initial pre-disaster
status, and all of the top four critical infrastructure systems (including electricity, roads,
telecommunications, and water supply) were disrupted and destroyed in the immediate aftermath of
Wenchuan Earthquake, which belonged to the extremely-high extent of damage. And the result shows
that the critical infrastructure systems are the most serious damage to this earthquake of all these four
dimensions, in large part due to the inadequate and aging infrastructure systems (Kathleen et al. 2010;
Whitman et al. 2013). As the initial pre-disaster status of critical infrastructure systems to be the
acceptable post-disaster level (black dotted line in Figure 11), the emergency critical infrastructure and
services was all restored just in three month (blue line in Figure 11): the emergency water supply and
telecommunications were recovered in the immediate time period, the emergency electricity in the
emergency time period, and the emergency roads in the short-run time period. The recovery score of




infrastructure $R_{infrastructure}$ is 135.19, which belongs to the high-recovery level, and is expected to be most
recovery compared with other three dimensions. The reliable and resilient infrastructure system is a
priority goal for earthquake-resilient communities, which is designed to continue functioning and
recover quickly within a shortly time period during and after earthquake disasters. Many researches
addressed the importance of enhancing defence infrastructure design to optimize mitigation, disaster
planning, and response and recovery efforts, which played a vital role in improving the community
recovery to earthquake disasters (Chang et al. 2011; National Infrastructure Advisory Council 2010).

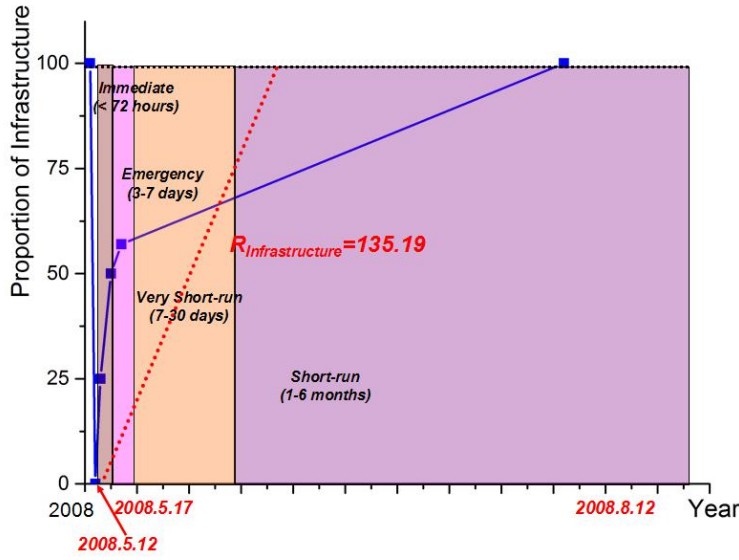

**Figure 11.** The recovery process and score of infrastructure of Wenchuan

**5 Discussion**

The overall results of our study highlight the community recovery process which is considered to be an
uncertain, complex, conflict-laden, multidimensional and nonlinear process. The extent of damage, land
use, building codes, available recovery resources, the broader structural changes, social disparities,
prevailing pre-disaster trends, decision making, and organization capacity are factors all directly related
to the rate of recovery. "Both long-term trends and an urgent desire to return to normal, exert an
important influence on the reconstruction processes" (Haas et al. 1977). And higher recovery scores
mean higher recovery levels and lower recovery scores mean lower recovery levels. The population,
building and infrastructure dimensions have high-recovery levels, particularly the infrastructure
recovery is highest. However, the economic recovery score is poor which tends to have lowest recovery
level in contrast to other three dimensions and needs more consideration in the near future. While the
external resources will be not sufficient to meet the needs of disaster-affected areas throughout the



recovery process of Wenchuan. The decision-makers of local government must learn how to address the
challenges of disaster response and recovery at the community level, how to leverage community
capacity from the earliest stages of disaster response, and to use external resources to bolster and
supplement local capacities. In the rebuilding and recovery process of Wenchuan, the community has
received a large number of external resources from Chinese Central Government and other provinces
and cities to enhance community recovery to earthquake, including incorporating long-term recovery
goals into disaster response and pre-disaster planning, expanding the knowledge base by incorporating
research into recovery and harnessing lessons learned from international experiences, and developing an
outcome-oriented approach to disaster recovery planning, which makes Wenchuan exhibit a high
recovery and the reconstructed community be more resilient to the next earthquake (Figure 2). The
rebuilding and recovery process of Wenchuan supports perspective of recent research that returning to
pre-disaster levels does not necessarily mean building back for the better (Ganapati et al. 2012). From a
dynamic and development oriented viewpoint, there is no exact returning to "pre-disaster" conditions
once a disaster has happened. Regardless of whether the disaster has stimulated positive change or has
hastened the development trend of a community, the community will never be exactly the same as it was
before the disaster occurred (Greene 2006). Furthermore, recovering to the pre-disaster situation implies
restoring the pre-event inequality, exploitation and vulnerability as well (Oliver-Smith 1990). The idea
of "build back better" (Lyons et al. 2010) or "recover better" should be adopted, especially in the case
of developing countries where "build back better" is indeed possible (Mulligan and Nadarajah 2012) if
the ideas of development, vulnerability and risk reduction are integrated into recovery activities (Shaw
2006), with the physical and social planning integrated with one another to address local needs in
culturally appropriate ways (Mulligan et al. 2012). And the post-disaster recovery activities provide an
opportunity to learn constantly and grow stronger from the previous natural disasters, which can be used
to support the proactive mitigation strategies-to rebuild stronger, change land-use patterns, and reduce
development in hazardous areas, and also to reshape those negative social, political, and economic
conditions that existed pre-event (NHC 2006; Reddy 2000; Olshansky 2006; Birkland 2006). Mitigation
can be a powerful tool for anticipating the unknown, for reducing losses, and for facilitating recovery
following a hazard impact. Mitigation strategies, for instance, may reduce potential losses by steering
development to the less hazardous areas of a proposed community or by modifying building design to
reduce potential losses (Burby et al. 1999). They are also useful in preparing communities to deal with
post-disaster scenarios by identifying actions that should be done prior to and immediately following
events to help guide recovery processes and to reduce future losses.
## 6 Conclusion
During the past few years a range of high profile, complex and uncertain earthquake disasters occurred
in China, such as the Wenchuan earthquake (May 12, 2008), the Yushu earthquake (April 14, 2010), and





the Ya'an earthquake (April 20, 2013), which have stimulated an escalation in theoretical developments
concerning the way to be quickly recovered from the earthquake damage. An examination of the current
and expected capabilities of communities to confront a potential shock yields understanding the
effective risk reduction strategies from another perspective, that build-in the resilient communities are
one of the key goals for emergency managers and decision makers to improve the local earthquake
prevention and response, and prioritize efforts that need to be undertaken in order to maximize the
effectiveness of various recovery measures. Effects to address these needs have focused upon new
approaches for analyzing the concept of community recovery and proposing community recovery
measurement methodologies. Thus, our research summarized some of the key themes emerging from
much of the current literature that defined a range of concepts of recovery, and proposed a new
perspective to identify the inherent characteristics of community recovery as the capacity to recover and
rebuild itself rapidly to an acceptable level of functioning and structure following the earthquake
disasters occur. By extending the recovery triangle, and on the basis of the principle of the equal area,
this paper developed a quantitative approach for measuring and characterizing the community recovery
to earthquake of Wenchuan in four dimensions (population, economy, building, and infrastructure). The
results suggest that most dimensions of Wenchuan represented the characteristics high recovery, while
infrastructure recovery is highest, and the economic recovery is lowest. The perspectives contributed to
understand the different recovery levels of different dimensions of Wenchuan for guiding planning of
appropriate response and reconstruction policies to enhance the community recovery to earthquake, and
emphasizing that the community recovery is strongly influenced by the decision making of local
governments. The measuring approach presents in this paper is intended to provide a quantitative
foundation for the future research of community recovery. It would be worthwhile conducting further
study to learn from the past recovery and rebuilding process for the development of appropriate
techniques of designing new mathematical models to measure and characterize community recovery,
which can help local government and policy makers develop the scientific and effective disaster
recovery plan for the next devastating earthquake disaster.

**Acknowledgments**

This work was supported by the National Natural Science Foundation of China under the project
No.71601042, the Humanity and Social Science Youth Foundation of Ministry of Education of China
under the project No.16YJC630071 and 16YJC630040, and China Postdoctoral Science Foundation
Funded Project No.2016M601401.

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
