# Peer review of "Measuring and Characterizing Community Recovery to Earthquake: the Case of"

_Natural Hazards and Earth System Sciences, 2017_

## Referee Comment (RC1) · Anonymous Referee #1 · 28 Feb 2017

I have read the paper entitled: "Measuring and characterizing community recovery to earthquake: the case of 2008 Wenchuan Earthquake, China" by Liu et al. The paper focuses on the very important topic of recovery and reconstruction of communities affected by natural hazards which has not been adequately investigated until now. Unfortunately, although the authors provide a well-written introduction presenting the problem and giving an overview of definitions and background, they fail in convincing the reader that the method they use is adequate to reach their goal. The interpretation of the results is also not thorough and is limited to a raw description. Although the final discussion includes some interesting and powerful statements, they are out of place or come too late in the manuscript. It seems to me that the discussion chapter

is less about the results and their meaning and more about these general statements. In more detail, I do not recommend the present article for publication in the NHESS Journal mainly due to the following concerns: 1. The authors develop a methodology for the "quantitative" measurement of community recovery. However, the method they present is not quantitative since the assessment or quantification of recovery is expressed through an abstract "score" and not a tangible value such as time, monetary value, number of people etc. 2. The authors do not demonstrate the usefulness and potential application of this method. Who are the potential end users and how can they make use of the method or the results? 3. The presentation of the method is rather confusing. Inconsistency in using some terms is contributing to this confusion. For example, in Figure 3 recovery is represented by curve. Later on in page 10 recovery equals the tan of angle a. Why "Area 1" equals "Area 2" in page 10, line 279? It is not clear which area is area 1 and which is area 2 in Figure 5. 4. The use of some terms have been also unclear throughout the text. According to the authors, (page 9, lines 259-267) extent of damage, robustness and system functionality is the same thing. ("Robustness (. . .) is considered to be the extent of damage of the community"). 5. Figures 7, 9, 10, 11 should be better explained: What is the dotted red line? What is the difference with the blue one? What are the blue dots? What do the colors mean (light blue, pink, grey etc.)? Why in some figures time starts before 2008? 6. The recovery scores presented in page 10, line 289 are rather unclear. What are the values in brackets? The tan of the angle? If this is the case how do you say that the $R_{population}=98.46$ belongs to the high recovery level? 7. What do you mean with maximum and minimum recovery (p.10, line 291)? Minimum recovery should be 0! 8. Figure 6 makes no sense to me. How can we put all the different dimensions together starting with a system function 0? System function is not the same for the different dimension. (By the way system infrastructure is missing from the figure) 9. More discussion about the X (extend of damage) is needed. How can this be expressed? If it is a percentage how can we measure the percentage of functionality loss of e.g. infrastructure? Is this extend of damage the same for each dimension (population,

economic, building, infrastructure)? Certainly not. If this is the case why do all Figures (7,9,10,11) start from system functionality 0? The dimensions are not comparable also because apart from the "extend of damage" also "time" is different. It is not possible to recover the buildings in one month, however, a successful recovery of population the way this is defined by the authors (recovery of injured people) should last less. 10. Why do the time in Figures 9 extends to 2018? Is there some prediction for the future in there that is not described thoroughly in the text? 11. Why do Figures 7, 9, 10, 11 start with from 0? Do you assume that from the time of the event the functionality fell so low? 12. Finally, the authors do not list their assumption and uncertainties related to their research. It would be a good idea to list all the assumptions or aspects that although they were important they were impossible to be implemented in the present research. They should also include some possible future developments of the present study.

Please also note the supplement to this comment:
http://www.nat-hazards-earth-syst-sci-discuss.net/nhess-2017-72/nhess-2017-72-RC1-supplement.pdf

---

## Referee Comment (RC2) · Anonymous Referee #2 · 28 Feb 2017

This paper presents a methodology to measure community recovery using linear projections of actual recovery rates and apply it to the earthquake stricken Wenchuan area in China. On-site assessments of earthquake recovery have potential to contribute to the literature, and so do methodological developments that improve our ability to measure recovery processes. Yet, the paper has some major flaws, particularly in the methodology, which I describe below.

General comments:

Section 1 The literature review is comprehensive, although I missed some key references on concept of economic resilience, which is cited a few times but not directly tackled. For example, dynamic economic resilience (Carter et al., 2007; Hallegatte et

al., 2016; Le De et al., 2013; Rose and Krausmann, 2013) is not explicitly presented, which is not what I would expect from a paper studying recovery. Authors may alternatively refer to "ability of the economy to cope, recover, and reconstruct and therefore to minimize aggregate consumption losses" (i.e. indirect impacts) (Hallegatte, 2014). References to indirect/dynamic resilience are necessary to put your contributions into context.

Section 2 Section 2 is sufficiently informative. In general, the manuscript would benefit from revision by a native English speaker.

Section 3 -Section 3.3 is not sufficiently detailed; methods are claimed to be a relevant contribution of the paper (and are in any case necessary to understand results) but are presented in less than 1 page. -The section also fails to put methods into context. The triangle approach is but one method to assess recovery, and this should be acknowledged. Alternatives should be presented, and pros and cons described (I wonder how relevant is this approach in the literature?). Authors should conclude stating why this method is used, and what their contribution offers as compared to other alternatives. -I have the impression that the triangle method may be too simplistic for the economic analysis part I'm familiar with. There is much work in this area relying on more complex models that could be applicable to the study area (see e.g. citations above). A reader familiar with these methods may wonder: why authors do not use them? It may be due to data constraints, or to keep homogeneity in the assessment of different recovery measures, or due to some limitations of the methods, but this has to be explained. -Authors provide some thresholds to assess recovery (immediate, emergency, etc.) and seem to apply them to every recovery measure (economic, population, etc.) without explaining the motives for this, and if this is coherent with the complexity of recovery and the different implications the concept has for the economy or infrastructures, for example. Overall, assessing these complex concepts with a single method seems challenging. -The definition of economic recovery in section 3.2 is insufficient. What's the counterfactual if you address GDP growth only? The original GDP growth rate? Do

you consider any trends? What about redistribution of income? -Section 3.2 seems an extension of the literature review in Section 1 and could be better placed there.

Section 4 This section needs to categorically discuss the results shown in the figures before moving on to assess the drivers. This would help readers to understand what we can obtain from the methods and how to interpret it.

Specific comments:

-P.1, l. 7-8: "So this article proposes the concept of community recovery as the capacity to recover and rebuild after the earthquake disasters by considering the original perspective of recovery." This sentence is critical to explain the reader how you intend to implement your analysis, but remains vague and imprecise. -P. 1, l.10: "by extending the concepts of recovery triangle". Here you should try to explain the methods employed in the paper in a way that even a reader that is not used to them understands how it's done. The current version is ok for a more focused journal, but in NHESS readers come from a variety of disciplines and papers must be informative for this audience. -P1, l.19: "The damaging earthquake risk of cities as the biggest risk of all natural disasters". This needs to be referenced, although earthquakes can be devastating there are other risks that happen more frequently. You can say that they are the most devastating in terms of impact, but not in terms of likelihood, and again this should be referenced. -P2, l. 37: possibilities to instead of "possibilities o return to normal" -P2 l.58: "Disaster Recovery Framework developed by FEMA in 2011(FEMA 2011)" say instead developed by FEMA (2011) to avoid repetition. -P3., l.100 to P.4, l.105. Again, methods are barely presented, which is not sufficient provided this is the main contribution of the paper. -P. 5, l. 139: "is from ruins to prosperity (Figure 2c)". I'd rather avoid bold statements like this. As you discussed before, rebuilding the city is just one part of the recovery. What about the human, natural, social capital that was lost, was it recovered? Has the city learned the lessons and is now more resilient to earthquakes? These and other questions need to be addressed before making this statement. -P.5, l. 151: "random interview of 1000 affected families". Why did you use this method, moti-

vate or else cite papers that used it, describe the method. -P.6, l.152: "Other statistics and description data are gathered by combining different sources (e.g. . . ..)". You have to describe categorically all databases used, or where you can find them, so that your methods can be replicated. There is no reference to the data sources, and the description is insufficient. Descriptive statistics could help. Otherwise a more in depth discussion of the data, its gaps, etc. is necessary. -P.10, l. 275: "For the purpose of facilitating the calculation, we use the average linear rate to substitute the curve rate." This sounds too simplistic and needs to be reinforced. Can't you estimate a non-linear function?

The list above is not exhaustive, and authors are advised to submit the document to a native English speaker or professional proofreading services.

Literature

Carter, M.R., Little, P.D., Mogues, T., Negatu, W., 2007. Poverty Traps and Natural Disasters in Ethiopia and Honduras. World Dev. 35, 835–856. doi:10.1016/j.worlddev.2006.09.010 Hallegatte, S., 2014. Economic Resilience: Definition and Measurement (Policy report), Policy Report Working Series. The World Bank. Hallegatte, S., Bangalore, M., Vogt-Schilb, A.C., 2016. Assessing socioeconomic resilience to floods in 90 countries (No. WPS7663). The World Bank. Le De, L., Gaillard, J.C., Friesen, W., 2013. Remittances and disaster: a review. Int. J. Disaster Risk Reduct. 4, 34–43. doi:10.1016/j.ijdrr.2013.03.007 Rose, A., Krausmann, E., 2013. An economic framework for the development of a resilience index for business recovery. Int. J. Disaster Risk Reduct. 5, 73–83. doi:10.1016/j.ijdrr.2013.08.003

---

## Short Comment (SC2) · 21 Mar 2017

This paper is an interesting and well researched contribution on measuring and characterizing Community Recovery to Earthquake. The authors provide a well-written introduction presenting the problem and giving an overview of definitions and background of community recovery, which is a very critical reference for other researchers. The authors defined the speed of the recovery process as the key indicator of measuring the community recovery, and measured the community by the recover rate is novel and valid, which has a great inspiration for the future research of community recovery. Therefore this work is very suitable for publication in NHESS after minor revision. For example, the English is not appropriate and can be thoroughly improved by the revision

of a native English speaker. The formal presentation of the manuscript has to fulfill the requirements of the format of the journal NHESS (i.e., references).

---

## Short Comment (SC3) · 10 Apr 2017

The paper describes a quantitative approach for measuring and characterizing the community recovery to earthquake. The paper is well structured, specially some perspectives the authors proposed in this paper have important and highly valuable efforts for the future research of community recovery. E.g. 'Earthquake-resilience community should be the new stable states and rebuilding a safer community in the historically experienced deleterious earthquake disasters' in Section 1 Introduction, 'The post-disaster recovery activities provide an opportunity to learn constantly and grow stronger from the previous natural disasters, which can be used to support the proactive mitigation strategies-to rebuild stronger, change land-use patterns, and reduce development

in hazardous areas, and also to reshape those negative social, political, and economic conditions that existed pre-event'in the Section 5 Discussion, and so on. This is an excellent manuscript and it was a pleasure to read. In conclusion it should be remarked this is a nicely structured, logical and sequential paper, which should certainly be published in NHESS. I only have a few minor comments for the section of the section 4 Results. This section is the main contribution of this paper to explain to our readers what to measure the community recovery of Wenchuan to Wenchuan Earthquake, how is the community recovery of Wenchuan and how to interpret and analyze the calculation results. So the authors should provide a more detailed analyses and discussion of the results shown in the figures 6-11.

---

## Author Comment (AC1) · 12 Apr 2017

Dear Reviewer:

Thanks very much for your comments concerning our manuscript entitled "Measuring and Characterizing Community Recovery to Earthquake: the Case of 2008 Wenchuan Earthquake, China" (ID: nhess-2017-72). Those comments are all valuable and very helpful for revising and improving our paper, as well as the important guiding significance to our researches. We have studied comments carefully and have made correction which we hope meet with approval. The revised paper has been uploaded in the supplement file, and the added and revised sentences and paragraphs have been marked red in the revised paper. The main corrections in the paper and the responds

to the reviewer's comments are as flowing:

We have revised and rewritten the section 3 Data and Methods to provide a clearer explanations of the methods which we proposed in this paper to assess the community recovery, that the readers can understand how to measure the community recovery in this paper. The revised sentences and paragraphs which are used to illustrate the assessment method of the community recovery have been added in the section 3.2 Defining and assessing the community recovery to earthquake (the page 8 to 13) of the revised paper, and marked red. For example, 1)Page 8, line 447-Page9, line 475: We provided a clearer explanations of the four properties of resilience (4R's) which was proposed by Bruneau et al., 2003. And the recover time which was from the term of rapidity in the four properties can be used by us to define and assess the community recovery. 2)Page 9, line 476-page 10, line 538: We illustrated the concept of resilience triangle (Figure 4) and the calculation formula of resilience (page 10, line 531). In the concept of resilience triangle, the recovery time is taken to assess community recovery. 3) Page 11,line 567-page 12, line 656: We provided a clearer explanations of the assessment method of the community recovery to earthquake. We have extended the original concept of resilience triangle and used the term of rapidity from four properties of resilience (4R's). So the speed at which the community recovers to achieve a desired state can be used in our paper to assess the community recovery. Figure 6 sketched the assessment framework of the community recovery we proposed in this section. And the recovery score (RS) is formulated as the following two-stage stochastic program from page 12, line 628 to page 12, line 656.

We have tried our best to improve the manuscript and made some changes in the manuscript. These changes will not influence the content and framework of the paper. And the revised manuscript has been typeset according to the format of NHESS. We appreciate for Editors (Thomas Thaler) and Reviewer's warm work earnestly, and hope that the correction will meet with approval. Once again, thanks very much for editors (Thomas Thaler) and the Reviewer's comments and suggestions

Please also note the supplement to this comment:
http://www.nat-hazards-earth-syst-sci-discuss.net/nhess-2017-72/nhess-2017-72-AC1-supplement.pdf

---

## Author Comment (AC2) · 12 Apr 2017

Dear Reviewer:

At first, we would like to express our great appreciation to you for your comments on our paper entitled "Measuring and Characterizing Community Recovery to Earthquake: the Case of 2008 Wenchuan Earthquake, China" (ID: nhess-2017-72). You have provided many valuable comments for us to modify our paper. These comments help us improve the quality of our paper, and made our paper more readable and clear. After reading these comments, we have realized the problems and deficiency of my manuscript, so we have revised my manuscript carefully and seriously according to your comments. The revised paper has been uploaded in the supplement file, and the

added and revised sentences and paragraphs have been marked red in the revised paper. The specific modifications of my manuscript are listed in the following:

1)Thank very much for your comments about the grammatical errors of our paper. Because English is not our native language, there are many spelling and grammatical errors in our paper. With the help of you and an English-native expert, numerous errors in grammar and syntax had been corrected, and the language of our manuscript had been improved. For example: Page 2, line 70-76: we have changed "So policymakers have called for concerted efforts to build 'earthquake-resilience community' for the purpose of finding the new stable states and rebuilding a safer community in the historically experienced deleterious earthquake disasters (Alesch 2009)." into "So policymakers have called for concerted efforts to build 'earthquake-resilience community' for the purpose to find the new stable states and rebuilding a safer community in the historically experienced deleterious earthquake disasters (Alesch, 2009)." Page 2, line 84-89: we have changed "Recovery represents a fundamental dimension of disaster resilience, includes both the possibilities o return to normal, that is, pre-disaster condition, and alternatively, to be rebuilt or transformed to a completely different status." into "Recovery represents a fundamental dimension of disaster resilience, includes both the possibilities to return to normal, that is, pre-disaster condition, or alternatively, to be rebuilt or transformed to a completely different status." Page 2, line 114-116: we have changed "since the disaster was often seen as a failure of social structure (Bates and Gillis Peacock 1989). " into ". . .since the disaster is often seen as a failure of social structure (Bates and Gillis Peacock, 1989)." and so on.

2) We have tried our best to improve the manuscript and made some changes in the manuscript. These changes will not influence the content and framework of the paper. And the revised manuscript has been typeset according to the format of NHESS, especially the references. For example, the text of our paper has been divided into two columns. And the formula has been numbered from (1) to (6). The first line of each paragraph has been indented by 2 characters. The format of references has been

checked and corrected as the format of NHESS.

We appreciate for Editors (Thomas Thaler) and Reviewer's warm work earnestly, and hope that the correction will meet with approval. Once again, thanks very much for editors (Thomas Thaler) and the Reviewer's comments and suggestions.

Please also note the supplement to this comment:
http://www.nat-hazards-earth-syst-sci-discuss.net/nhess-2017-72/nhess-2017-72-AC2-supplement.pdf

---

## Author Comment (AC3) · 12 Apr 2017

Dear Reviewer:

Thanks very much for your comments concerning our manuscript entitled "Measuring and Characterizing Community Recovery to Earthquake: the Case of 2008 Wenchuan Earthquake, China" (ID: nhess-2017-72). Those comments are all valuable and very helpful for revising and improving our paper, as well as the important guiding significance to our researches. We have studied comments carefully and have made correction which we hope meet with approval. The revised paper has been uploaded in the supplement file, and the added and revised sentences and paragraphs have been marked red in the revised paper. The main corrections in the paper and the responds

to the reviewer's comments are as flowing:

We are very sorry that we can't provide a clear explanations of the assessment results of Wenchuan's recovery to earthquake. According to the reviewer's comments, in the revised paper, we have redrawn the figures (Figure 7, 8, 9, 10, 11), and rewritten the section 4 Results to help readers to understand what we can obtain from the methods and how to interpret it. For example: (1)We have provided a explanation of the interrelated phases of the recovery and reconstruction process(page 14, line 768-802). That we have divided the recovery and reconstruction process into three interrelated phases (shown in Figure 7), which can be used to determine the recovery degrees of four dimensions of community recovery at different time phases .

(2) We have moved the concept of 4 dimensions of community recovery respectively into 4.1-4.4. And we have added a lot of explanation of the assessment results of each dimension of Wenchuan's recovery to earthquake, which has been marked red (page 15, line 814-page 20, line 1151). And all of the Figures (figure 8-11) have been redrawn. And each figure which has shown the assessment result of the four dimensions (population, economy, building and infrastructure) of Wenchuan recovery has been illustrated in detail in the section 4.1 (page 15, line 814-page 16, line 893), 4.2 (page 16, line 898-page 17, line 1002), 4.3 (page 18, line 1007-1078), 4.4 (page 19, line 1082-page 20, line 1148), and marked red in these sections. We wish these revision can provide a clearer interpretation of the assessment results.

We have tried our best to improve the manuscript and made some changes in the manuscript. These changes will not influence the content and framework of the paper. And the revised manuscript has been typeset according to the format of NHESS. We appreciate for Editors (Thomas Thaler) and Reviewer's warm work earnestly, and hope that the correction will meet with approval. Once again, thanks very much for editors (Thomas Thaler) and the Reviewer's comments and suggestions.

Please also note the supplement to this comment:

http://www.nat-hazards-earth-syst-sci-discuss.net/nhess-2017-72/nhess-2017-72-AC3-supplement.pdf

---

## Author Comment (AC4) · 12 Apr 2017

Dear Referee:

At first, we would like to express our great appreciation to you for your comments on our paper entitled "Measuring and Characterizing Community Recovery to Earthquake: the Case of 2008 Wenchuan Earthquake, China" (ID: nhess-2017-72). Those comments are all valuable and very helpful for revising and improving the quality of our paper, made our paper more readable and clear, as well as the important guiding significance to our researches. We have studied comments carefully and realized the problems and deficiency of my manuscript, so we have revised our manuscript carefully and seriously according to your comments, which we hope meet with approval. The revised paper

has been uploaded in the supplement file, and the added and revised sentences and paragraphs have been marked red in the revised paper. The main corrections in the paper and the responds to your comments are as following:

1.Comments: The authors develop a methodology for the "quantitative" measurement of community recovery. However, the method they present is not quantitative since the assessment or quantification of recovery is expressed through an abstract "score" and not a tangible value such as time, monetary value, number of people etc.

Answer: As the referee said, in our paper, we have presented a framework for defining community recovery and using the indicator of recovery score which can be expressed by the value of recovery speed to assess it. So in the revised paper, we have replaced "measure", "measurement" or "measuring" with "assess", "assessment" or "assessing". We wish this term can provide a clearer interpretation of the quantitative measurement we have proposed in this paper.

2.Comments: The authors do not demonstrate the usefulness and potential application of this method. Who are the potential end users and how can they make use of the method or the results?

Answer: According to the referee's comments, we have added the purpose of our paper section 1 introduction (page 4, line 236-271). We have illustrated that "After 2008 Wenchuan Earthquake, Chinese Central Government have provided disaster assistance and developed many recovery programs for the impacted communities. The total investment of these recovery programs is 1 trillion yuan. The local government officials take the most important role in the post-disaster recovery. So when these emergency response activities and programs carried out, challenges must be faced and key decisions made included of Chinese Central Government is to assess the recovery capacity and performance." "Wenchuan Earthquake provides an important opportunity to learn from the decisions made by the local governments and their consequences for recovery. So the intended outcome of this paper is to propose a new,

practical method for assessing and characterizing community recovery to earthquake in four dimensions, and applied it to Wenchuan Community. The final products of our research provide insights for Chinese Central Government to assess and measure the recovery capacity and performance of local government officials of Wenchuan, in order to maximize the overall post-disaster community recovery by prioritizing efforts, and formulating effective, operational and valuable reconstruction strategies and policies in the future." So the usefulness and potential application of this method is to provide insights for Chinese Central Government to assess and measure the recovery capacity and performance of local government officials of Wenchuan. Especially we have added the section 3.3 Core dimensions and indicators of community recovery (page 13, line 660-page 14, line 733) to provide a clearer explanation of why we use these dimensions and indicators to assess the community recovery. That is "performing the core dimensions and indicators of community recovery, it is necessary to answer the question the community recovery "of what" and "to what" should be the most concerned by Chinese Central Government. In addition, the choice of the core dimensions and indicators of community recovery depends on the particular case (Wenchuan) for assessment, as well as on availability of data". And "according to the characteristics of earthquake disaster, and in order to better interpret all aspects of community recovery, a total of 15 interviews involving 20 experts were conducted to judge and choose the core dimensions and indicators of community recovery, which can significantly reflect local government capacity the recovery capacity and performance of local government officials. All of these experts were organizational specialists on post-disaster recovery and reconstruction from National Workplace Emergency Management Center which can be the decision-makers of assessing and measuring the recovery capacity and performance of local government officials". Furthermore "core dimensions and indicators of community recovery was defined and choose on the basis of three stages: first, the dimensions was developed from a systematic analysis of existing recovery assessment literature, which gathered together a set of qualitative indicators of community recovery; and second, that the expert interview collectively represented the

entire dimensions and indicators for the experts to judge the most important core indicators of each dimension. Last, we captured and summarized the experts judgments of the core dimensions and indicators of community recovery. That four core indicators were chose to assess the four dimensions of community recovery, which included: (a) population recovery, assessed by the recovered quality of the interviewed affected families; (b) economy recovery, assessed by the recovered quality of gross domestic product (GDP); (c) building recovery, assessed by the recovered quality of damaged or destroyed buildings, and (d) infrastructure recovery, assessed by the recovered quality of key infrastructure system (e.g. electricity, roads, telecommunications, and water supply)."

3.Comments: The presentation of the method is rather confusing. Inconsistency in using some terms is contributing to this confusion. For example, in Figure 3 recovery is represented by curve. Later on in page 10 recovery equals the tan of angle a. Why "Area 1" equals "Area 2" in page 10, line 279? It is not clear which area is area 1 and which is area 2 in Figure 5.

Answer: We are very sorry that we can't provide a clear explanations of the definition and assessment method of the community recovery. So in the revised paper, we have rewritten the section 3 Data and Methods and redrawn the figures (Figure 4, 5, 6) so that the readers can understand how to measure the community recovery in this paper. The added and revised sentences and paragraphs which used to illustrate the assessment method of the community recovery have been mainly in the section 3.2 Defining and assessing the community recovery to earthquake (the page 8 to 13) of the revised paper, and marked red. For example, 1) Page 8, line 447-Page9, line 475: We provided a clearer explanations of the four properties of resilience (4R's) which proposed by Bruneau et al., 2003. And the recover time which was from the term of rapidity in the four properties was always used to define and assess the community recovery before. 2) Page 9, line 476-page 10, line 538: We illustrated the concept of resilience triangle (Figure 4) and the calculation formula of resilience (page 10, line 531). In the concept

of resilience triangle, the recovery time is taken to assess community recovery. 3) Page 11,line 567-page 12, line 656: We provided a clearer explanations of the assessment method of the community recovery to earthquake. That is we extended the original concept of resilience triangle and used the term of rapidity from four properties of resilience (4R's). So the speed at which the community recovers to achieve a desired state can be used in our paper to assess the community recovery. Figure 6 sketched the assessment framework of the community recovery we proposed in this section. And the recovery score (RS) is formulated as the following two-stage stochastic program from page 12, line 628 to page 12, line 656. 4) We have illustrated the reason why we use the parameter of recovery speed to assess the community recovery. And the advantages of using the parameter of recovery speed, the disadvantages of using the parameter of recovery time in 'resilience triangle' which is the classic parameters to assess community recovery have been illustrated in page 10, line 539-page 11, line 562.

4.Comments: The use of some terms have been also unclear throughout the text. According to the authors, (page 9, lines 259-267) extent of damage, robustness and system functionality is the same thing. ("Robustness (. . .) is considered to be the extent of damage of the community").

Answer: We are very sorry that we can't provide a clearer explanation of some terms we used in this paper. So according to the referee's comments, in the revised paper, we have redrawn the figures (Figure 4, 5, 6) in section 3.2 Defining and assessing the community recovery to earthquake (the page 8 to 13), and illustrated these terms in detail. For example: 1)We have added the explanation of the four properties of resilience (4R's) proposed by Bruneau et al. (2003) including robustness, redundancy, resourcefulness, and rapidity in page 8, line 447-page 9, line 475. 2)We have added the explanation of the definition of resilience by using "resilience triangle" in page 9, line 476-page 10, line 538. And figure 4 have illustrated the concept of resilience triangle. Some key terms have been expressed in detail, such as Q(t) has been defined for

the percentage 'functionality' (or 'quality', or 'serviceability') of a community. And t is time, an earthquake occurs at time t0, the community is completely repaired at time t1. R has been the loss of community resilience. 3)We have added the explanation of the assessing process of the community recovery in page 12, line 625-656. That the recovery score has been formulated as the following two-stage stochastic program. And figure 6 has sketched the assessment framework of the community recovery. Some key terms have been expressed in detail, such as Rcurve is the loss of resilience experienced by the community in the curve functionality recovery path; Rlinear is the loss of resilience experienced by the community in the linear functionality recovery path; RS is recovery score that can be expressed by the value of recovery speed; Q(t)curve is the percentage functionality of the community in the curve functionality recovery path; Q(t)linear is the percentage functionality of the community in the linear functionality recovery path; Q(t0)linear is the percentage functionality of the community at the time of earthquake occurrence in the linear functionality recovery path; t0 is the time instant when the earthquake occurs; t1 is the length of recover time in the curve functionality recovery path; t2 is the length of recover time in the linear functionality recovery path;is the tangent angle of the linear functionality recovery path.

5.Comments: Figures 7, 9, 10, 11 should be better explained: What is the dotted red line? What is the difference with the blue one? What are the blue dots? What do the colors mean (light blue, pink, grey etc.)? Why in some figures time starts before 2008?

Answer: We are very sorry that we can't provide a clear explanations of the assessment results of Wenchuan's recovery to earthquake. According to the referee's comments, in the revised paper, we have redrawn the figures (Figure 7, 8, 9, 10, 11), and rewritten the section 4 Results to help readers to understand what we can obtain from the methods and how to interpret it. 1)The meaning of the colors in these figures has been illustrated in page 14, line 768-line 802. That is "according to the time phases of community recovery proposed by Rubin(1985), National Research Council (2011) and FEMA, we divided the recovery and reconstruction process into three interrelated phases (shown

in Figure 7), which can be used to determine the recovery degree of four dimensions of community recovery at different time phases: short-term (marked red), intermediate recovery(marked yellow), and long-term recovery (marked blue)". 2)The meaning of the black curve, red dotted line and other lines has been illustrated in the 4 dimensions of community recovery respectively (section 4.1-4.4). Such as the black curve plotted in the figures have shown the actual recovered process of the 4 dimensions (population, economy, building and infrastructure) in months following the earthquake disaster. Red dotted line plotted in these figures have shown the approximate recovered process of the 4 dimensions (population, economy, building and infrastructure), which is calculated by the assessment method we proposed in 3.2. The x-axis in these figures has shown the months following the earthquake disaster. The y-axis in these figures has shown the recovered quality of the 4 dimensions (percent).

6.Comments: The recovery scores presented in page 10, line 289 are rather unclear. What are the values in brackets? The tan of the angle? If this is the case how do you say that the Rpopulation=98.46 belongs to the high recovery level?

Answer: We are very sorry that we can't provide a clear explanations of the assessment results of Wenchuan's recovery to earthquake. So in the revised paper, we have rewritten the section 4 Results to help readers to understand what we can obtain from the methods and how to interpret it. For example: 1) We have provided a explanation of the interrelated phases of the recovery and reconstruction process(page 14, line 768-802). That we have divided the recovery and reconstruction process into three interrelated phases (shown in Figure 7), which can be used to determine the recovery degree of four dimensions of community recovery at different time phases . 2) We have moved the concept of 4 dimensions of community recovery respectively into 4.1-4.4. And we have added a lot of explanation of the assessment results of each dimension of Wenchuan's recovery to earthquake, which has been marked red (page 15, line 814-page 20, line 1151). And all of the Figures (figure 8-11) have been redrawn. We wish these revision can provide a clearer interpretation of the assessment results. 3) We

have provided a clearer explanation of the recovery levels in page 14, line 743-767. In the revised paper, "three levels (low-recovery, medium-recovery, high-recovery) with the recovery scores are adopted in this study to assess the degree of recovery. So the low-recovery level belongs to the calculation of the recovery score RS as [0-0.577] and the tangent angle $\alpha$ as [0°-30°], the medium-recovery level belongs to the calculation of the recovery score RS as [0.577-1.732] and the tangent angle $\alpha$ as (30°-60°), the high-recovery level belongs to the calculation of the recovery score RS as [1.732-+$\infty$] and the tangent angle $\alpha$ as (60°-90°). The calculation results suggest that the economic recovery which can be obtained by the recovery score RSeconomy=1.15 is the minimum value in the four dimensions, and the infrastructure recovery which can be obtained by the recovery score RSinfrastructure=135.19 is maximum value in the four dimensions. And the economic recovery of Wenhuan which belongs to the medium-recovery level, the population, buildings and infrastructure recovery belong to the high-recovery level".

7.Comments: What do you mean with maximum and minimum recovery (p.10, line 291)? Minimum recovery should be 0!

Answer: We are very sorry we can't provide a clearer explanation of the maximum and minimum recovery. We have provided a clearer explanation of the maximum and minimum recovery in page 14, line 756-763. That is "The calculation results suggest that the economic recovery which can be obtained by the recovery score RSeconomy=1.15 is the minimum value in the four dimensions, and the infrastructure recovery which can be obtained by the recovery score RSinfrastructure=135.19 is maximum value in the four dimensions".

8.Comments: Figure 6 makes no sense to me. How can we put all the different dimensions together starting with a system function 0? System function is not the same for the different dimension. (By the way system infrastructure is missing from the figure)

Answer: According to the referee's comments, we also think the figure 6 makes no sense to this paper. So we have deleted this figure in the revised paper.

9.Comments: More discussion about the X (extend of damage) is needed. How can this be expressed? If it is a percentage how can we measure the percentage of functionality loss of e.g. infrastructure? Is this extend of damage the same for each dimension (population, economic, building, infrastructure)? Certainly not. If this is the case why do all Figures (7,9,10,11) start from system functionality 0? The dimensions are not comparable also because apart from the "extend of damage" also "time" is different. It is not possible to recover the buildings in one month, however, a successful recovery of population the way this is defined by the authors (recovery of injured people) should last less.

Answer: 1) We are very sorry that we can't provide a clearer explanation of the figures (figure 8-11) in the section 4 Results. So all of the figures (figure 8-11) have been re-drawn. And each figure which has shown the assessment result of the four dimensions (population, economy, building and infrastructure) of Wenchuan recovery has been illustrated in detail in the section 4.1 (page 15, line 814-page 16, line 893), 4.2 (page 16, line 898-page 17, line 1002), 4.3 (page 18, line 1007-1078), 4.4 (page 19, line 1082-page 20, line 1148), and marked red in these sections. We wish these revision can provide a clearer interpretation of the assessment results. 2) The meaning of the x-axis has been from the "resilience triangle" proposed by Bruneau et al. (2003), which we have clearly illustrated its definition in page 9, line 476-page 10, line 538. And figure 4 have illustrated the concept of resilience triangle. In figure 4, the x-axis has been the quality of system, which is Q(t) that been defined for the percentage 'functionality' (or 'quality', or 'serviceability') of a community. And the y-axis has been the time, that the earthquake occurs at time t0, the community is completely repaired at time t1. In our assessment method of community recovery, we have extended the original concept of resilience triangle and use the term of rapidity from four properties of resilience (4R's) (Bruneau et al., 2003) to assess community recovery, which refers to how fast the community returns towards equilibrium after the earthquake. So in the analysis of assessment results(section 4.1-4.4), the meaning of the terms used in the figures have been illustrated, including the x-axis in these figures has shown the months following

the earthquake disaster, the y-axis in these figures has shown the recovered quality of the 4 dimensions (percent), RS is recovery score of the 4 dimensions that can be expressed by the value of recovery speed, is the tangent angle of of the 4 dimensions of the linear functionality recovery path.

10.Comments: Why do the time in Figures 9 extends to 2018? Is there some prediction for the future in there that is not described thoroughly in the text?

Answer: We are very sorry that we can't provide a clear explanations of the assessment results of Wenchuan's recovery to earthquake. So in the revised paper, we have rewritten the section 4 Results to help readers to understand what we can obtain from the methods and how to interpret it. Figures 9 (page 17, line 1004) have been redrawn. The reason why the time in Figure 9 extends to 2018 has been illustrated in page 16, line 909-page 17, line 971. That "the statistical data of the gross domestic product (GDP) which we have used to assess the economy recovery is in the period from 2008-2016. But until 2016, statistical data shows that Wenchuan's GDP did not attain pre-disaster levels, which briefly recovered to 60 percent of the entire quality. So we assumes that the GDP after 2016 increases as the average growth rate (25.2%) of 8 years after the earthquake (2008-2016), and finally it recovered to the pre-disaster level in 2018".

11.Comments: Why do Figures 7, 9, 10, 11 start with from 0? Do you assume that from the time of the event the functionality fell so low?

Answer: 1)We are very sorry that we can't provide a clearer explanation of the figures (figure 8-11) in the section 4 Results. So all of the figures (figure 8-11) have been redrawn. And each figure which has shown the assessment result of the four dimensions (population, economy, building and infrastructure) of Wenchuan recovery has been illustrated in detail in the section 4.1 (page 15, line 814-page 16, line 893), 4.2 (page 16, line 898-page 17, line 1002), 4.3 (page 18, line 1007-page 19, line 1078), 4.4 (page 19, line 1082-1148), and marked red in these sections. We wish these revision can

provide a clearer interpretation of the assessment results. 2) In the revised section 4.1-4.4, the meaning of the terms used in the figures have been illustrated, including the x-axis in these figures has shown the months following the earthquake disaster, the y-axis in these figures has shown the recovered quality of the 4 dimensions (percent), RS is recovery score of the 4 dimensions that can be expressed by the value of recovery speed, is the tangent angle of of the 4 dimensions of the linear functionality recovery path. The data of damage extent of each dimension after the Wenchuan Earthquake were obtained from the reports on the work of the Wenchuan government from 2008 to 2016, and gathered by combining different sources (e.g., research report, government report, government agency and website) following the Wenchuan Earthquake. These data sources has been explained clearly in the section 3.1 Data Sources (page 6, line 348-page 8, line 414). Owing to the huge destructive earthquake, four dimensions (population, economy, building and infrastructure) of Wenchuan has been seriously damaged. For example, after the Wechuan Earthquake occurred, more than 80% families and population were severely injured, even homeless (page 15, line 847-850). And after the Wechuan Earthquake occurred, the GDP of Wenchuan is only 22.53% of the pre-disaster status (page 16, line 919-921). And the status of buildings of Wenchuan before the earthquake disaster can be set as the initial pre-disaster status, and more than 90 percent of these buildings were damaged even destroyed in Wenchuan Earthquake (page 18, line 1020-1025), and so on.

12. Comments: Finally, the authors do not list their assumption and uncertainties related to their research. It would be a good idea to list all the assumptions or aspects that although they were important they were impossible to be implemented in the present research. They should also include some possible future developments of the present study.

Answer: According to the referee's comments, we have added the limitations and our future research in the section 6 conclusion (page 23, line 1323-page 24, line 1391). For example: 1)The limitations of our paper including "First, the approach is focused

on one specific earthquake scenario (Wenchuan Earthquake) and one specific community (Wenchuan). Consequently, variations in effects across other potential earthquakes and other characteristics of communities were not discussed". "Second, assessing community recovery is focused on describing core dimensions and indicators which can used by the decision-makers to assess and measure the recovery capacity and performance of local government officials (for example, identifying GDP to assess economy recovery), not considering other economic or social indicators, such as personal income, poverty, and unemployment, and so on, in assessing patterns and progress of community recovery". "Third, the statistical data used to assess different dimensions of community recovery are likely to be sparser and less reliable, special surveys or arrangements with data collecting authorities may then be necessary in the future research". "Last, core indicators of community recovery was defined and chose on the basis of expert interview, these experts we interviewed are all from one organization (National Workplace Emergency Management Center), who may not always have a complete understanding of community recovery". 2)Our future research including "Quantitative indicators of community recovery should be used as a benchmark or reference for more in-depth study, which can be used systematically by local governments and researchers to monitor complex recovery processes". "Validation may be possible in the future through expanded databases of the consequences of earthquakes for comparable regions, in order to give the operator a wider and deeper insight in the recovery patterns of different communities". "The concept framework of community recovery should be evaluated and revised more efficiently and effectively by collecting and analyzing a large number of expert judgments". "Considering long-term recovery and reconstruction, the framework should be extended in order to perform a dynamic assessment model of community recovery, where time-dependent indicators reflect post-disaster recovery capacity and performance of local government officials over time". "Learning from the past recovery and rebuilding process, new research is needed to fully operationalize and realize the concept of recovery, and develop appropriate techniques of designing mathematical models to assess and characterize

community recovery, which can help local government and policy makers develop the scientific and effective disaster recovery plan for the next devastating earthquake disaster."

We have tried our best to improve the manuscript and made some changes in the manuscript. These changes will not influence the content and framework of the paper. And the revised manuscript has been typeset according to the format of NHESS. We appreciate for Editors (Thomas Thaler) and referee's warm work earnestly, and hope that the correction will meet with approval. Once again, thanks very much for editors (Thomas Thaler) and the referee's comments and suggestions.

Please also note the supplement to this comment:
http://www.nat-hazards-earth-syst-sci-discuss.net/nhess-2017-72/nhess-2017-72-AC4-supplement.pdf

---

## Author Comment (AC5) · 12 Apr 2017

Dear Referee:

At first, we would like to express our great appreciation to you for your comments on our paper entitled "Measuring and Characterizing Community Recovery to Earthquake: the Case of 2008 Wenchuan Earthquake, China" (ID: nhess-2017-72). Those comments are all valuable and very helpful for revising and improving the quality of our paper, made our paper more readable and clear, as well as the important guiding significance to our researches. We have studied comments carefully and realized the problems and deficiency of my manuscript, so we have revised my manuscript carefully and seriously according to your comments, which we hope meet with approval. The revised paper

has been uploaded in the supplement file, and the added and revised sentences and paragraphs have been marked red in the revised paper. The main corrections in the paper and the responds to your comments are as following:

1.Section 1 Comments: The literature review is comprehensive, although I missed some key references on concept of economic resilience, which is cited a few times but not directly tackled. For example, dynamic economic resilience (Carter et al., 2007; Hallegatte et al., 2016; Le De et al., 2013; Rose and Krausmann, 2013) is not explicitly presented, which is not what I would expect from a paper studying recovery. Authors may alternatively refer to "ability of the economy to cope, recover, and reconstruct and therefore to minimize aggregate consumption losses" (i.e. indirect impacts) (Hallegatte, 2014). References to indirect/dynamic resilience are necessary to put your contributions into context.

Answer: The referee provided an important reference (Hallegatte, 2014) for us to research economic recovery. So in Page 3, line 182-186, we have replaced "dynamic economic resilience (Carter et al., 2007; Hallegatte et al., 2016; Le De et al., 2013; Rose and Krausmann, 2013)" with "ability of the economy to cope, recover, and reconstruct and therefore to minimize aggregate consumption losses (i.e. indirect impacts) by Hallegatte (2014). And we have added this reference "Hallegatte, S.: Risk and Opportunity–Managing Risk for Development, World Development Report , Washington, DC, World Bank, 2014." in the section of References (Page 25, line 1472-1475).

2.Section 2 Comments: Section 2 is sufficiently informative. In general, the manuscript would benefit from revision by a native English speaker.

Answer: Thank very much for your comments about the grammatical errors of our paper. Because English is not our native language, there are many spelling and grammatical errors in our paper. With the help of you and an English-native expert, numerous errors in grammar and syntax had been corrected, and the language of our manuscript had been improved.

3.Section 3 Comments: Section 3.3 is not sufficiently detailed; methods are claimed to be a relevant contribution of the paper (and are in any case necessary to understand results) but are presented in less than 1 page. -The section also fails to put methods into context. The triangle approach is but one method to assess recovery, and this should be acknowledged. Alternatives should be presented, and pros and cons described (I wonder how relevant is this approach in the literature?). Authors should conclude stating why this method is used, and what their contribution offers as compared to other alternatives. -I have the impression that the triangle method may be too simplistic for the economic analysis part I'm familiar with. There is much work in this area relying on more complex models that could be applicable to the study area (see e.g. citations above). A reader familiar with these methods may wonder: why authors do not use them? It may be due to data constraints, or to keep homogeneity in the assessment of different recovery measures, or due to some limitations of the methods, but this has to be explained. -Authors provide some thresholds to assess recovery (immediate, emergency, etc.) and seem to apply them to every recovery measure (economic, population, etc.) without explaining the motives for this, and if this is coherent with the complexity of recovery and the different implications the concept has for the economy or infrastructures, for example. Overall, assessing these complex concepts with a single method seems challenging. -The definition of economic recovery in section 3.2 is insufficient. What's the counterfactual if you address GDP growth only? The original GDP growth rate? Do you consider any trends? What about redistribution of income? -Section 3.2 seems an extension of the literature review in Section 1 and could be better placed there.

Answer: (1) We are very sorry that we can't provide a clear explanations of the definition and assessment method of the community recovery. So in the revised paper, we have rewritten the section 3 Data and Methods so that the reader can understand how to measure the community recovery in this paper. The added and revised sentences and paragraphs which used to illustrate the assessment method of the community recovery have been mainly in the section 3.2 Defining and assessing the community

recovery to earthquake (the page 8 to 13) of the revised paper, and marked red. For example, 1) Page 8, line 447-Page9, line 475: We provided a clearer explanations of the four properties of resilience (4R's) which proposed by Bruneau et al., 2003. And the recover time which was from the term of rapidity in the four properties was always used to define and assess the community recovery before. 2) Page 9, line 476-page 10, line 538: We illustrated the concept of resilience triangle (Figure 4) and the calculation formula of resilience (page 10, line 531). In the concept of resilience triangle, the recovery time is taken to assess community recovery. 3) Page 11,line 567-page 12, line 656: We provided a clearer explanations of the assessment method of the community recovery to earthquake. That is we extended the original concept of resilience triangle and used the term of rapidity from four properties of resilience (4R's). So the speed at which the community recovers to achieve a desired state can be used in our paper to assess the community recovery. Figure 6 sketched the assessment framework of the community recovery we proposed in this section. And the recovery score (RS) is formulated as the following two-stage stochastic program from page 12, line 628 to page 12, line 656. (2) We have illustrated the reason why we use the parameter of recovery speed to assess the community recovery, and the advantages of using the parameter, and the disadvantages of using the parameter of recovery time in 'resilience triangle' which is the classic parameters to assess community recovery are in page 10, line 539-page 11, line 562. (3) The motive and purpose of our paper is to provide insights for Chinese Central Government to assess and measure the recovery capacity and performance of local government officials of Wenchuan, which have been illustrated in page 4, line 236-line 271. That "the core dimensions and indicators which we used in this paper to assess the community recovery have been judged and chosen by a total of 15 interviews involving 20 experts. All of these experts were organizational specialists on post-disaster recovery and reconstruction from National Workplace Emergency Management Center which can be the decision-makers of assessing and measuring the recovery capacity and performance of local government officials". We have added the section 3.3 Core dimensions and indicators of community recovery (page 13, line

660-page 14, line 733) to illustrate it in detail. (4) As the referee said, the core dimensions and indicators (population, economy, building and infrastructure) we have chosen and used to assess the community recovery was not comprehensive. That is the limitation of our paper, which have been illustrated in section 6 Conclusion (page 23, line 1323-1356). For example, "assessing community recovery is focused on describing core dimensions and indicators which can used by the decision-makers to assess and measure the recovery capacity and performance of local government officials (for example, identifying GDP to assess economy recovery), not considering other economic or social indicators, such as personal income, poverty, and unemployment, and so on, in assessing patterns and progress of community recovery", and "core indicators of community recovery was defined and chose on the basis of expert interview, these experts we interviewed are all from one organization (National Workplace Emergency Management Center), who may not always have a complete understanding of community recovery".

4.Section 4 Comments: This section needs to categorically discuss the results shown in the figures before moving on to assess the drivers. This would help readers to understand what we can obtain from the methods and how to interpret it.

Answer: We are very sorry that we can't provide a clear explanations of the assessment results of Wenchuan's recovery to earthquake. So in the revised paper, we have rewritten the section 4 Results to help readers to understand what we can obtain from the methods and how to interpret it. For example: (1) We have provided a explanation of the interrelated phases of the recovery and reconstruction process(page 14, line 768-802). That we have divided the recovery and reconstruction process into three interrelated phases (shown in Figure 7), which can be used to determine the recovery degree of four dimensions of community recovery at different time phases . (2) We have moved the concept of 4 dimensions of community recovery respectively into 4.1-4.4. And we have added a lot of explanation of the assessment results of each dimension of Wenchuan's recovery to earthquake, which has been marked red (page 15,

line 814-page 20, line 1151). And all of the Figures (figure 8-11) have been redrawn. We wish these revision can provide a clearer interpretation of the assessment results.

5. Specific comments:

(1) P.1, l. 7-8: "So this article proposes the concept of community recovery as the capacity to recover and rebuild after the earthquake disasters by considering the original perspective of recovery." This sentence is critical to explain the reader how you intend to implement your analysis, but remains vague and imprecise.

Answer: Page 1, line 12-16: According to the referee's comments, we have changed the sentence "So this article proposes the concept of community recovery as the capacity to recover and rebuild after the earthquake disasters by considering the original perspective of recovery." into "Considering the original perspective of recovery, this article proposes the concept of community recovery as the capacity to recover and rebuild after the earthquake disasters by considering the original perspective of recovery."

(2) P. 1, l.10: "by extending the concepts of recovery triangle". Here you should try to explain the methods employed in the paper in a way that even a reader that is not used to them understands how it's done. The current version is ok for a more focused journal, but in NHESS readers come from a variety of disciplines and papers must be informative for this audience.

Answer: 1) Page 8, line 447-Page9, line 475: We provided a clearer explanations of the four properties of resilience (4R's) which proposed by Bruneau et al., 2003. And the recover time which was from the term of rapidity in the four properties can be used by us to define and assess the community recovery. 2) Page 9, line 476-page 10, line 538: We illustrated the concept of resilience triangle (Figure 4) and the calculation formula of resilience (page 10, line 531). In the concept of resilience triangle, the recovery time is taken to assess community recovery.

(3) P1, l.19: "The damaging earthquake risk of cities as the biggest risk of all natural

disasters". This needs to be referenced, although earthquakes can be devastating there are other risks that happen more frequently. You can say that they are the most devastating in terms of impact, but not in terms of likelihood, and again this should be referenced.

Answer: Page 1, line 42-44: According to the referee's comments, we have changed the sentence "The damaging earthquake risk of cities as the biggest risk of all natural disasters" into "The damaging earthquake risk of cities as the most devastating in terms of impact, but not in terms of likelihood. . .".

(4) P2, l. 37: possibilities to instead of "possibilities o return to normal"

Answer: Page 2, line 86-89: According to the referee's comments, we have changed the sentence ". . .includes both the possibilities o return to normal, that is, pre-disaster condition, and alternatively, to be rebuilt or transformed to a completely different status." into ". . .includes both the possibilities to return to normal, that is, pre-disaster condition, or alternatively, to be rebuilt or transformed to a completely different status."

(5) P2, l.58: "Disaster Recovery Framework developed by FEMA in 2011(FEMA 2011)" say instead developed by FEMA (2011) to avoid repetition.

Answer: Page 3, line 134-136: According to the referee's comments, we have changed the sentence " The new National Disaster Recovery Framework developed by FEMA in 2011(FEMA 2011) define recovery as. . ." into "The new National Disaster Recovery Framework developed by FEMA (2011) define recovery as. . ."

(6) P3., l.100 to P.4, l.105. Again, methods are barely presented, which is not sufficient provided this is the main contribution of the paper.

Answer: In the revised paper, we have rewritten the section 3 Data and Methods so that the reader can understand how to measure the community recovery in this paper. The added and revised sentences and paragraphs which used to illustrate the assessment method of the community recovery have been mainly in the section 3.2 Defining and

assessing the community recovery to earthquake (the page 8 to 13) of the revised paper, and marked red.

(7)P. 5, l. 139: "is from ruins to prosperity (Figure 2c)". I'd rather avoid bold statements like this.

Answer: Page 6, line 340: According to the referee's comments, we have deleted the sentence "the Wenchuan Community is from ruins to prosperity".

(8)As you discussed before, rebuilding the city is just one part of the recovery. What about the human, natural, social capital that was lost, was it recovered? Has the city learned the lessons and is now more resilient to earthquakes? These and other questions need to be addressed before making this statement.

Answer: 1) In the section 5 discussion, we have illustrated what we have learned from this research of the Wenchuan recovery to earthquake. That is "the decision-makers of local government must learn how to address the challenges of disaster response and recovery at the community level, how to leverage community capacity from the earliest stages of disaster response, and to use external resources to bolster and supplement local capacities (Page 21, line 1181-1188)". And "The rebuilding and recovery process of Wenchuan supports perspective of recent research that returning to pre-disaster levels does not necessarily mean building back for the better (Ganapati et al., 2012) (Page 21, line 1204-1208)". Furthermore "And the post-disaster recovery activities provide an opportunity to learn constantly and grow stronger from the previous natural disasters, which can be used to support the proactive mitigation strategies-to rebuild stronger, change land-use patterns, and reduce development in hazardous areas, and also to reshape those negative social, political, and economic conditions that existed pre-event (NHC, 2006; Reddy, 2000; Olshansky, 2006; Birkland, 2006) (page 21, line 1231-page 22, line 1242)", and so on. 2) Because the motive and purpose of our paper is to provide insights for Chinese Central Government to assess and measure the recovery capacity and performance of local government officials of Wenchuan. In this paper, we

just proposed a assessment method of the community recovery in 4 core dimensions, which have been judged and chosen by a total of 15 interviews involving 20 experts. All of these experts were organizational specialists on post-disaster recovery and reconstruction from National Workplace Emergency Management Center which can be the decision-makers of assessing and measuring the recovery capacity and performance of local government officials. The research of the community recovery in this paper can not consider other dimensions (such as the human, natural, social capital, and so on). So in page 23, line 1323-page 24, line 1391, we have added the limitation and our future research. For example, we have illustrated that "Second, assessing community recovery is focused on describing core dimensions and indicators which can used by the decision-makers to assess and measure the recovery capacity and performance of local government officials (for example, identifying GDP to assess economy recovery), not considering other economic or social indicators, such as personal income, poverty, and unemployment, and so on, in assessing patterns and progress of community recovery(Page 23, line 1334-1344)". And "Last, core indicators of community recovery was defined and chose on the basis of expert interview, these experts we interviewed are all from one organization (National Workplace Emergency Management Center), who may not always have a complete understanding of community recovery (Page 23, line 1350-1356)". In our future research, we have illustrated that "quantitative indicators of community recovery should be used as a benchmark or reference for more in-depth study, which can be used systematically by local governments and researchers to monitor complex recovery processes (Page 23, line 1360-1365)". And "And considering long-term recovery and reconstruction, the framework should be extended in order to perform a dynamic assessment model of community recovery, where time-dependent indicators reflect post-disaster recovery capacity and performance of local government officials over time (Page 23, line 1375-1382)". And "Learning from the past recovery and rebuilding process, new research is needed to fully operationalize and realize the concept of recovery, and develop appropriate techniques of designing mathematical models to assess and characterize community recovery, which can help local government and policy makers develop the scientific and effective disaster recovery plan for the next devastating earthquake disaster (Page 23, line 1382-page 24, line 1391)".

(9)P.5, l.151: "random interview of 1000 affected families". Why did you use this method, motivate or else cite papers that used it, describe the method.

Answer: Page 6, line 355-375: We have illustrated questionnaire and interview process of the affected families, and how to get the data of the recovery process and status of the affected people.

(10)-P.6, l.152: "Other statistics and description data are gathered by combining different sources (e.g. . ..)". You have to describe categorically all databases used, or where you can find them, so that your methods can be replicated. There is no reference to the data sources, and the description is insufficient. Descriptive statistics could help. Otherwise a more in depth discussion of the data, its gaps, etc. is necessary.

Answer: Page 7, line 381-385, and Table 1: We have provided a clearer explanation of the statistics and description data sources. we have illustrated that "Other statistics and description data (showed in table 1) are gathered by combining different sources (e.g., research report, government report, government agency and website) following the Wenchuan Earthquake." And Table 1 has listed the statistics and description data sources.

(11) P.10, l. 275: "For the purpose of facilitating the calculation, we use the average linear rate to substitute the curve rate." This sounds too simplistic and needs to be reinforced. Can't you estimate a non-linear function?

Answer: We are very sorry that we can't provide a clear explanations of calculation method of the community recovery. So in the revised paper, we have rewritten calculation process in the section 3.2 Defining and assessing the community recovery to earthquake (the page 11, line 567 to Page 12, line 656). The speed at which the community recovers to achieve a desired state can be used in our paper to assess

the community recovery. Figure 6 (Page 13) sketches the assessment framework of community recovery. But quantifying the slope of the recovery curve to assess the community recovery is very difficult and a challenge in this paper, because the recovery speed of the curve is different at each time point, and not a constant. For the purpose of facilitating the calculation, assuming that the performance of community of the resilience is unchanged and equal, we use the linear functionality recovery path to approximate the curve functionality recovery path. the recovery score is formulated as the following two-stage stochastic program in page 12, line 628-656.

(12)The list above is not exhaustive, and authors are advised to submit the document to a native English speaker or professional proofreading services.

Answer: Thank very much for your comments about the grammatical errors of our paper. Because English is not our native language, there are many spelling and grammatical errors in our paper. With the help of you and an English-native expert, numerous errors in grammar and syntax had been corrected, and the language of our manuscript had been improved. For example: 1) Page 2, line 70-76: we have changed "So policymakers have called for concerted efforts to build 'earthquake-resilience community' for the purpose of finding the new stable states and rebuilding a safer community in the historically experienced deleterious earthquake disasters (Alesch 2009)." into "So policymakers have called for concerted efforts to build 'earthquake-resilience community' for the purpose to find the new stable states and rebuilding a safer community in the historically experienced deleterious earthquake disasters (Alesch, 2009)." 2) Page 2, line 84-89: we have changed "Recovery represents a fundamental dimension of disaster resilience, includes both the possibilities o return to normal, that is, pre-disaster condition, and alternatively, to be rebuilt or transformed to a completely different status." into "Recovery represents a fundamental dimension of disaster resilience, includes both the possibilities to return to normal, that is, pre-disaster condition, or alternatively, to be rebuilt or transformed to a completely different status." 3) Page 2, line 114-116: we have changed "...since the disaster was often seen as a failure of social structure

(Bates and Gillis Peacock 1989). " into "...since the disaster is often seen as a failure of social structure (Bates and Gillis Peacock, 1989)." and so on.

We have tried our best to improve the manuscript and made some changes in the manuscript. These changes will not influence the content and framework of the paper. And the revised manuscript has been typeset according to the format of NHESS. We appreciate for Editors (Thomas Thaler) and Reviewers' warm work earnestly, and hope that the correction will meet with approval. Once again, thanks very much for editors (Thomas Thaler) and four Reviewers' comments and suggestions.

Please also note the supplement to this comment:
http://www.nat-hazards-earth-syst-sci-discuss.net/nhess-2017-72/nhess-2017-72-AC5-supplement.pdf